# Phosphoregulation of the novel hemi-arrestin MAPK scaffold Sms1 prevents untimely mating

Boris Sieber ✉, Laura Merlini, Wanlan Li, Maëlys Besomi, Laetitia Michon, Sushila Gordon-Lennox & Sophie G. Martin ✉

Mitogen-activated protein kinases (MAPK) are ancestral kinases that form essential signalling cascades. However, scaffolds that recruit kinases to subcellular locations and promote signal transduction have only been described in a few species. Notably, no scaffold was thought necessary for the MAPK cascade promoting sexual differentiation in fission yeast. Here, we identify the hemi-arrestin protein Sms1 as a novel scaffold of this MAPK cascade. Interactions with PIP2 and the pheromone receptor–coupled Gα subunit target Sms1 to plasma membrane patches, where it assembles the active cascade by binding each MAP kinase. These interactions are essential for signal transduction and local signal interpretation for polarised growth. Phosphorylation, including by the MAPK itself, antagonises Sms1 membrane translocation, establishing a negative feedback that underlies polarity patch turnover and prevents untimely mating attempts. Thus, Sms1 is a MAPK scaffold with canonical functions despite its distinct structural fold, highlighting convergent evolution of MAPK scaffolds across eukaryotes.

MAPK signalling, one of the best characterised transduction pathways across evolution, induces a wide range of responses ranging from cell proliferation to cell differentiation and sexual reproduction[1]. The core of the pathway is formed by a three-tier kinase cascade wherein the MAP3K phosphorylates the MAP2K, which in turn phosphorylates the MAPK. These three kinases are widely conserved in all fungi, plants and mammals[2–4] and likely already existed in the last eukaryotic common ancestor[5]. As a single MAPK cascade converts distinct upstream stimuli into context-dependent outputs, a critical question is how kinase activity is spatiotemporally regulated and how substrate specificity is determined.

Key factors required for MAP kinase activation and substrate specificity are the MAPK scaffolds. MAPK scaffolds, which can be defined as proteins that interact with at least two MAP kinases, ensure signalling at specific, subcellular localisations by locally increasing kinases concentration and thus activity[6,7]. The best characterised scaffolds of ERK-like MAPK include the budding yeast protein Ste5 and the metazoan proteins KSR1/2 and β-arrestins (Fig. 1A). Ste5 was first

discovered and is essential for pheromone signalling[8–11]. KSR1/2 is critical for Ras-induced tumorigenesis[12]. β-arrestins, beyond their canonical roles in binding phosphorylated G-protein coupled receptors (GPCR) to arrest signalling[13], also scaffold MAPK signalling for cytosolic ERK activation[14–16]. Interestingly, despite their divergent structures, these MAPK scaffolds display the same three critical features. First, they bind all three MAP kinases[8,14,15,17–21] and trigger allosteric kinase activation[22–27]. Through these interactions, scaffolds ensure transduction specificity, for instance, directing signalling from a MAP3K shared between three cascades to a specific MAPK output in the case of Ste5 or from G-protein-driven GPCR signalling towards MAPK signalling in the case of β-arrestins[7,28]. Second, they localise and recruit the MAPK cascade to the cell surface by direct interaction with phospholipids and with membrane-localised factors, such as the active GPCR or the released Gβγ[15,17,22,24,29–34]. Finally, phosphorylation of the MAPK scaffold, in part promoted by MAPK-dependent feedback, negatively regulates the scaffold membrane localisation, preventing untimely activation or terminating signalling[18,21,35–43].

Department of Molecular and Cellular Biology, University of Geneva, Geneva, Switzerland. ✉e-mail: Boris.Sieber@unige.ch; Sophie.Martin@unige.ch

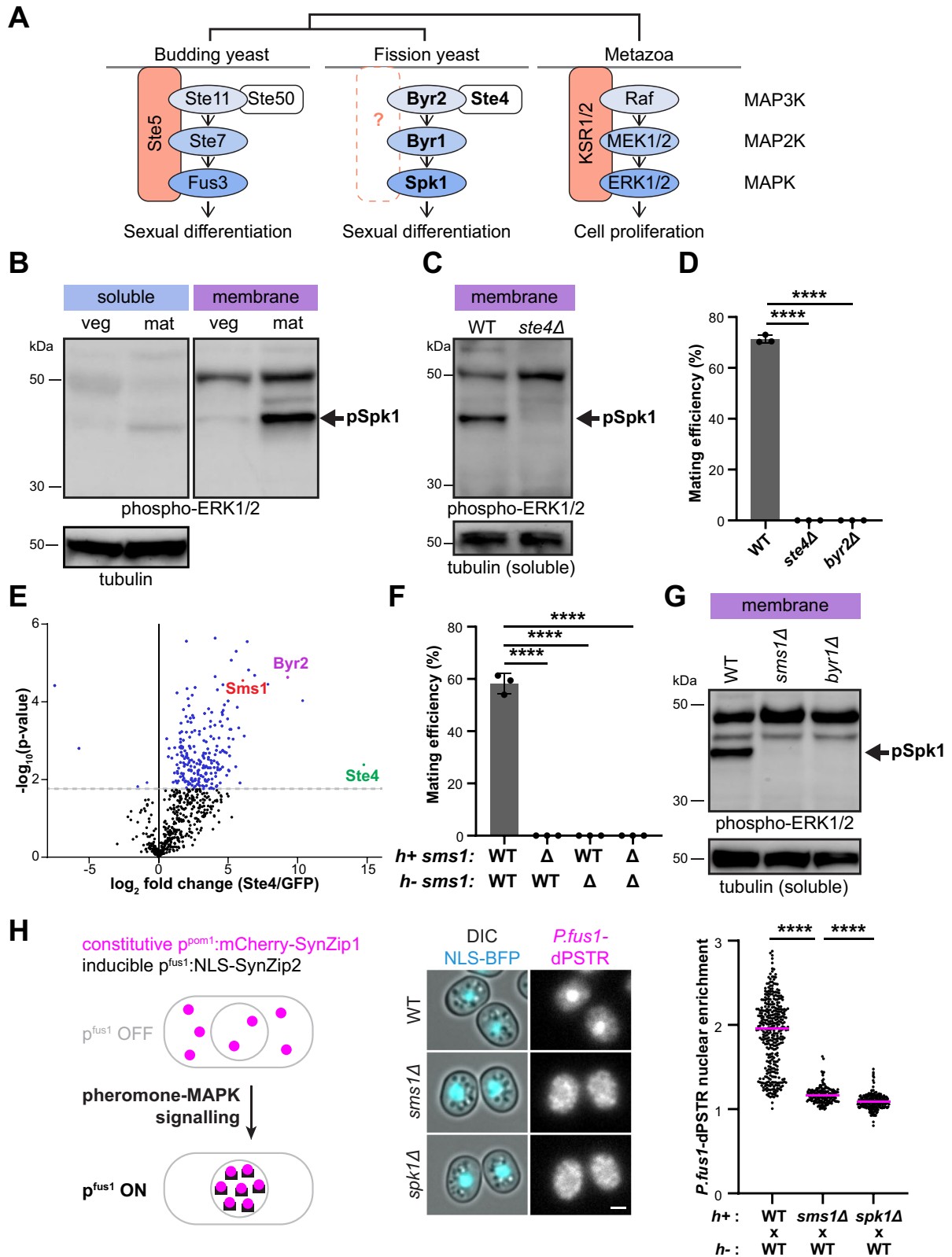

In contrast to the highly conserved and ancient MAPK signalling cascade[5] and despite having critical roles, MAPK scaffolds do not appear to be evolutionarily conserved. For instance, KSR and β-arrestins are restricted to Metazoa[5,44]. Similarly, Ste5 only exists in *Saccharomycotina*, where it is thought to have arisen at the same time as MAPK duplication[45]. This raises the questions of whether scaffolds are only required in some organisms and how MAPK signalling is efficiently transduced in others. For instance, although MAPK signalling plays an essential role in transducing pheromone signalling during fission yeast mating and can be partially replaced by human or budding yeast orthologs[46–48], no scaffold has been identified. Given the absence of shared kinases between different MAPK cascades in this organism[45], we—and others—have considered for more than 30 years that no MAPK scaffold was involved in fission yeast mating (Fig. 1A)[45,49–53].

**Fig. 1 | Sms1 is a novel, essential component of pheromone signalling in fission yeast. A** Schematic of MAPK signalling and scaffolds. **B** Phospho-Spk1 (arrow) in membrane and soluble fractions from *h90* proliferating vegetative cells in MSL + N (veg) and mating cells in MSL-N (mat). Tubulin serves as loading control. Note that the membrane fraction blot is also shown in Fig. 5C. **C** Phospho-Spk1 (arrow) in membrane fractions from *h90* WT and *ste4Δ* cells in MSL-N. **D** Mating efficiency of *h90* WT, *ste4Δ* and *byr2Δ* cells ($n \geq 500$ cells for three independent experiments) with error bars as s.d. ****$P < 0.0001$, one-way ANOVA with Tukey test. **E** Volcano plot representing the results of mass-spectrometry analysis from three Ste4-GFP co-immunoprecipitations in mating cells. The grey dotted line indicates p-values < 0.05, two-tailed *t*-test with Benjamini-Hochberg FDR correction. **F** Mating efficiency of heterothallic WT and *sms1Δ* cells ($n \geq 500$ cells for three independent experiments) with error bars as s.d. ****$P < 0.0001$, one-way ANOVA with Tukey test. **G** Phospho-Spk1 (arrow) in membrane fractions from WT, *sms1Δ* and *byr1Δ* cells in MSL-N. **H** dPSTR analysis of MAPK^Spk1 transcriptional output. Images and quantification of nuclear enrichment of the dPSTR reporter in h + WT, *sms1Δ* and *spk1Δ* cells crossed with *h-* WT after 24 h on MSL-N. Unmated cells were chosen for analysis. $n > 190$ cells for two independent experiments. Scale bar: 2 μm. ****$P < 0.0001$, Kruskal-Wallis with Dunn's test.

Pheromone-triggered MAPK signalling governs sexual reproduction in fission yeast and is temporally strictly controlled. Pheromones, expressed by each mating type, bind cognate GPCRs on the partner cell, leading to activation of the coupled Gα Gpa1 and signal transmission through a MAPK cascade that comprises the Ras GTPase-binding MAP3K Byr2, the MAP2K Byr1 and the MAPK Spk1 (Fig. 1A)[48,50]. Byr2 further binds an essential adaptor Ste4, which acts in parallel to Ras1-GTP, upstream of the MAP3K[54,55]. Sexual differentiation is suppressed during cell proliferation in rich environments and initiated by nitrogen starvation, which induces the derepression of the transcription factor Ste11 that controls the expression of most pheromone cascade components[56,57]. Ste11 is itself a MAPK substrate and its transcriptional activity is further enhanced by pheromone-MAPK signalling, forming a positive feedback loop that drives the cells into the sexual differentiation programme[58]. Differentiation underlies the fusion of mating partners to form a diploid zygote, in which the Mi-Pi bipartite transcription factor is assembled and triggers the activation of the RNA-binding protein Mei2. In turn, Mei2 signalling represses mating and initiates meiosis to form 4 haploid spore progenies[59,60].

In addition to driving transcription, the pheromone signal is also interpreted spatially. Throughout mating, subcellular compartmentalisation of the information is crucial to ensure directional pheromone response resulting in partner selection and cell-cell fusion. Pheromone secretion and reception occur in small, restricted polarity patches that transiently form at the plasma membrane at mating onset[61,62], a behaviour conserved in budding yeast[63–66]. These transient patches, which contain the active form of Cdc42 GTPase, as well as the Gα Gpa1 and active Ras1-GTP, undergo cycles of assembly and disassembly until stabilised by increasing pheromone signalling that drives partner cells to grow towards each other[62,67,68]. Patch proteins, as well as all MAP kinases, further localise at the mature polarity patch, where they promote cell-cell fusion to form the zygote[69]. How the activated receptor transduces the pheromone signal and recruits the MAPK cascade to the plasma membrane during fission yeast mating remains an open question.

Here, we discover Sms1, which was previously described as essential for mating and inhibitory for meiosis and known as Ste7 ref. [70], as the essential scaffold of the MAPK signalling pathway in fission yeast mating. Sms1 is recruited to the plasma membrane by its hemi-arrestin domain and restricted to pheromone signalling patches by specific recognition of the Gα. We show that Sms1 is essential to assemble an active MAPK cascade at the membrane by binding the MAP3K through the adaptor Ste4, as well as the MAP2K and MAPK. Interestingly, Sms1 is required even in the presence of hyperactive MAP2K, demonstrating a role beyond kinase activation. Finally, we discovered that Sms1 is negatively regulated by phosphorylation, in part through the MAPK, to prevent pheromone hypersensitivity and untimely mating attempts. The shared features of Sms1 with other MAPK scaffolds—membrane recruitment, MAPK scaffolding, signal transduction and negative feedback regulation—but absence of sequence or structural homology demonstrates convergent evolution of the MAPK scaffolds.

## Results

### Sms1 is a novel, essential component of pheromone signalling in fission yeast

To verify that pheromone-MAPK signalling occurs at the plasma membrane in fission yeast, we first checked if the active pool of MAPK^Spk1 localises at the membrane. To this aim, we performed membrane extraction[71,72], which we validated by verifying enrichment of the membrane-associated receptor-coupled prenylated Gα^Gpa1 protein (Fig. S1A). Activated, phospho-MAPK^Spk1 was detected with the cross-reacting phospho-ERK antibody against the conserved TEY motif in ERK-like MAP kinases (Fig. S1B)[73]. This revealed that phosphorylated MAPK^Spk1 is enriched in the membrane fraction in mating cells (Fig. 1B), demonstrating activation of the ERK-like MAPK^Spk1 at the membrane.

To identify novel interactors of the MAPK cascade, we focused on the MAP3K^Byr2 adaptor Ste4, which belongs to the Ste50 family conserved throughout fungi[74,75]. Ste4 is essential for the mating process as its deletion abrogates Spk1 phosphorylation and prevents mating (Fig. 1C, D)[76]. To identify novel Ste4 interactors, we immunoprecipitated endogenous Ste4-GFP or GFP as a negative control from mating cell lysates and analysed the bead eluate by mass spectrometry. We identified 235 significantly enriched proteins, including the MAP3K^Byr2, thus validating the experimental design (Fig. 1E and Table S1). Amongst Ste4 interactors, we focused on the poorly characterised protein SPAC23E2.03c (UniprotKB Q10136), which was previously identified in sterile screens and known as Ste7[70]. This protein was shown to be a critical component of cell conjugation that represses meiosis onset, likely through interaction with Mei2 and its kinase inhibitor Pat1 and is unstable in zygotes[70]. To avoid any confusion with the homonym MAP2K Ste7 in *S. cerevisiae* and upon approval by the PomBase Gene Naming committee, we changed its name to Sms1 (scaffold of MAPK signalling for reasons below).

Deletion of *sms1* resulted in sterility, even when mated with a wildtype partner (Fig. 1F), consistent with a previous report[70]. This phenotype is shared by loss of components of the pheromone-MAPK signalling pathway[77,78]. In agreement with the idea that Sms1 is a core component of MAPK signalling, loss of Sms1 almost fully abrogated MAPK phosphorylation (Fig. 1G). To probe its function in MAPK-dependent transcriptional response, we used the dynamic protein synthesis translocation reporter (dPSTR) assay, previously established in budding yeast[79] (Fig. 1H). In the dPSTR assay, nuclear translocation of a constitutively expressed mCherry-SynZip1, which tightly binds a pheromone-induced *P.fus1*:NLS-SynZip2, provides a dynamic measure of gene expression. Nuclear translocation in either mating type was abrogated in *spk1Δ* cells, demonstrating that the assay specifically reports on MAPK-dependent transcriptional response (Figs. 1H and S1C). In *sms1Δ*, nuclear translocation was strongly reduced. Thus, Sms1 is an essential component for the transduction of the pheromone-MAPK signal.

### Sms1 interacts with the MAP3K adaptor Ste4 via β-sheet augmentation

To verify the interaction of Sms1 with Ste4, we performed co-immunoprecipitation of Sms1 and Ste4, which we expressed in mitotic cells (Fig. 2A). As this experiment was performed in proliferating

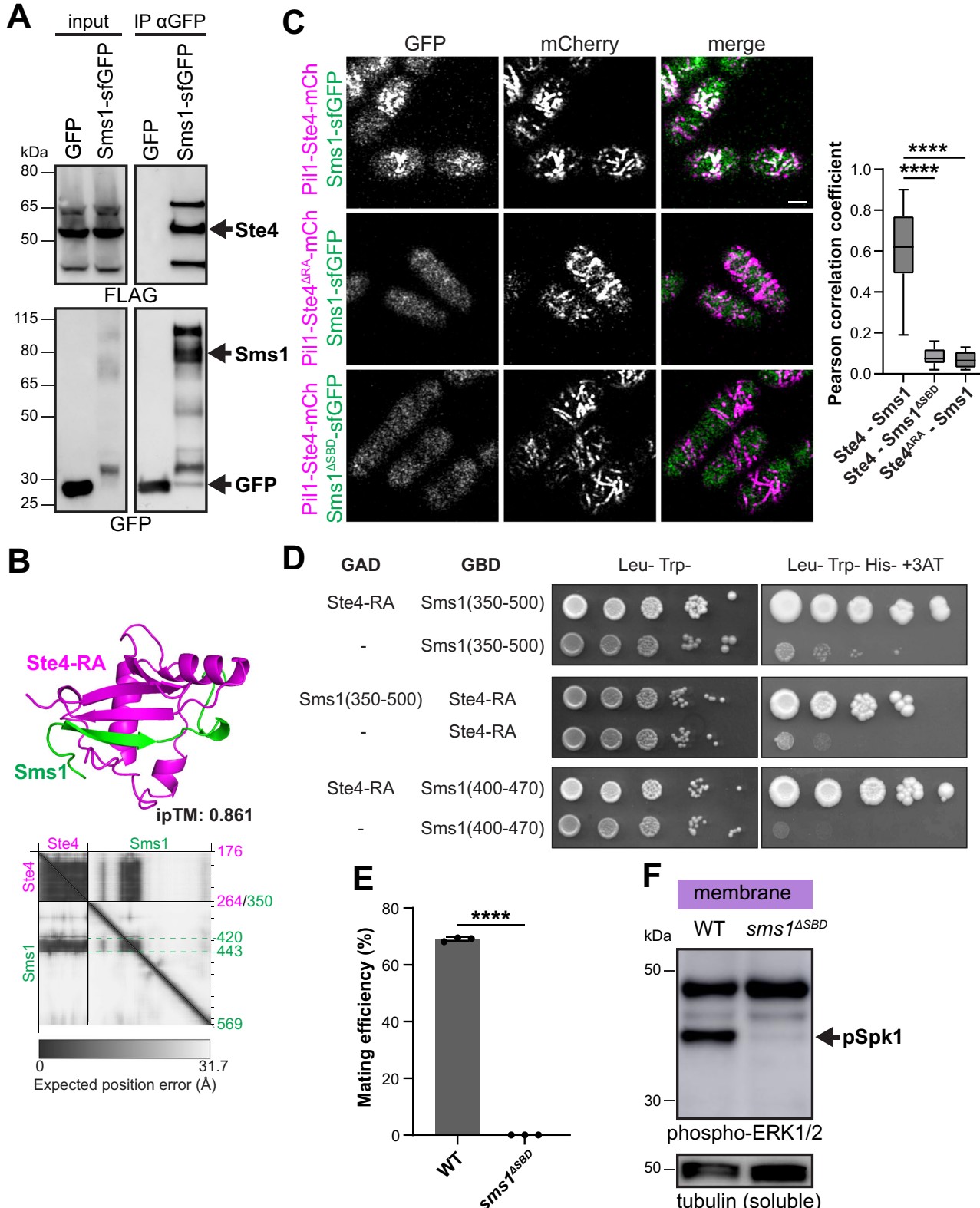

cells that lack Ste11-dependent transcription of the mating-specific proteins[57,80,81], this result indicates that the interaction between Sms1 and Ste4 does not require other pheromone-induced factors and may be direct.

Ste4 N-terminus binds the MAP3K[Byr2] via SAM-SAM domain interaction, an interaction abolished in the SAM mutant Byr2[N28I] (Fig. S2A)[54,55,82]. AlphaFold-based predictions indicated that the C-terminal Ras-associated (RA) domain of Ste4 would interact with

aa 422-433 of Sms1 via β-sheet augmentation (Fig. 2B)[83,84]. To test this prediction, we used a co-recruitment assay where an mCherry-tag bait is fused to Pil1 to target it to eisosomes, long invaginations in the plasma membrane of fission yeast and recruitment of a GFP-tagged prey is assessed by microscopy[85]. As proof of principle, Pil1-Ste4 recruited MAP3K[Byr2] and co-localisation was abolished by the N28I mutation in MAP3K[Byr2], but not by truncation of the Ste4 RA domain (Fig. S2B). Consistent with our co-immunoprecipitation

**Fig. 2 | Sms1 interacts with the MAP3K cofactor Ste4 via β-sheet augmentation.**
**A** Ste4-3xFLAG-mCherry co-precipitates with Sms1-sfGFP. Cell lysates from vegetative cells expressing Ste4-3xFLAG-mCherry and Sms1-sfGFP or GFP under *nmt41* promoter were immunoprecipitated with anti-GFP beads and immunoblotted for FLAG or GFP. GFP serves as negative control. **B** AlphaFold-Multimer (v2) prediction of the interaction between Ste4 RA domain and Sms1, showing Sms1 residues within 3 Å of Ste4 (aa 420-443). The predicted aligned error (PAE) plot and interface-predicted template modelling (ipTM) score are shown. **C** Colocalisation of Pil1-Ste4-mCherry (WT or ΔRA) with Sms1-sfGFP (WT or ΔSBD). Scale bar: 2 µm. Pearson correlation coefficient values are shown on the right ($n \geq 5$ cells for three independent experiments) with boxplots, where the central line indicates the median, the bottom and top edges of the box indicate the 25th and 75th percentiles and the whiskers extend to the smallest and largest values. ****$P < 0.0001$, Kruskal-Wallis with Dunn's test. **D** Yeast-two-hybrid assay showing that Sms1(400-470) is sufficient for interaction with Ste4-RA. **E** Mating efficiency of *h90* WT (*sms1-sfGFP*) and *sms1^ΔSBD^-sfGFP* cells ($n \geq 500$ cells for three independent experiments) with error bars as s.d. ****$P < 0.0001$, two-tailed Student *t* test. **F** Phospho-Spk1 (arrow) in membrane fractions from WT (*sms1-sfGFP*) and *sms1^ΔSBD^-sfGFP* in MSL-N.

results, Pil1-Ste4 strongly recruited Sms1 to eisosomes (Fig. 2C). Truncation of the Ste4 RA domain abolished the interaction with Sms1, as did the replacement of the 12-amino acid long β-strand in Sms1 (*sms1^ΔSBD^*; Fig. 2C). To probe the sufficiency of the predicted regions for Ste4-Sms1 interaction, we further performed yeast two-hybrid assay which confirmed the Ste4 binding domain of Sms1 to the 400-470 amino acid region (Fig. 2D). Interestingly, AlphaFold only predicts a β-strand in this region upon interaction with Ste4 and not in monomeric Sms1 (Fig. 2B). This interaction therefore represents an example of coupled folding of an intrinsically disordered region[86,87]. Taken together, these results are consistent with the AlphaFold structural prediction of interaction through β-sheet augmentation.

To test the physiological relevance of the Ste4-Sms1 interaction, we deleted the Ste4 RA domain or the Sms1 Ste4-binding domain (SBD) from the respective endogenous genes. Cells with *ste4^ΔRA^* had nearly absent phospho-Spk1 signal and were fully sterile, indicating this domain is indeed essential for function (Fig. S2C, D). The *sms1^ΔSBD^* allele similarly showed a strong reduction in Spk1 phosphorylation and prevented mating, confirming the importance of Sms1-Ste4 interaction for function (Fig. 2E, F).

## Sms1 localises to membrane-associated polarity patches and fusion sites

Tagging of Sms1 with sfGFP at the endogenous locus confirmed that Sms1 is expressed upon nitrogen starvation and its expression is strongly increased by pheromone stimulation (Fig. S3A). As Ste11 is de-repressed in these conditions, this is consistent with Ste11-dependent transcription of *sms1*[57,70,88]. It also revealed that Sms1-sfGFP localises in discrete membrane-associated patches in mating cells before accumulating at the point of contact between the two mating partners (Fig. 3A), where the actin fusion focus assembles to drive cell-cell fusion[89]. The membrane-associated Sms1 patches in mating cells colocalised with the Cdc42 scaffold Scd2, indicating that Sms1 is present in mobile polarity patches prior to mating partner selection (Fig. 3B), like Ras1 and Gα^Gpa1 62,67. Sms1 localisation mirrored that of Ste4, which also localised to mobile polarity patches (Fig. 3C, D). Upon cell-cell fusion, Sms1 dissociated from its membrane-bound domains and was cleared from the zygote (Fig. S3B), consistent with a previous report[70]. A decrease in protein level upon cell-cell fusion was not observed for Ste4 (Fig. S3B), suggesting a distinct regulation of Sms1 in the zygote. Together, these results identify Sms1 as a novel, essential member of the pheromone signalling pathway that polarises to the cell surface during the mating process.

Immunoblotting after whole-cell extraction of mating cells expressing Sms1-sfGFP only showed low molecular weight fragments, in contrast to the strong full-length band of Ste4-GFP, suggesting that Sms1 is an unstable protein (Fig. S3C). Immunoprecipitation of Sms1-sfGFP fragments showed that the protein is ubiquitinated (Fig. S3D). Optimisation of the extraction buffer and protocol (see 'methods') allowed to recover some full-length Sms1, though a large fraction was still degraded (Fig. 3E, soluble fraction). Similarly, Sms1 over-expression in mitotic cells showed some full-length product, but a large fraction was also degraded (Fig. S3E). By contrast, expression of Sms1 in the hypomorphic *mts3-1* proteasome mutant[90] strongly stabilised the full-length band of Sms1 (Fig. S3E), demonstrating that Sms1 is degraded by the ubiquitin-proteasome system.

Since Sms1 is unstable yet essential for mating, we hypothesised that Sms1 might be differentially regulated at the membrane (polarity patches and fusion site) versus the cytosol. Indeed, membrane extraction of endogenous Sms1-sfGFP revealed more abundant full-length protein, in contrast with the low-molecular-weight degradation fragments present in the soluble fraction (Fig. 3E). High-molecular-weight bands further suggested post-translational modifications of Sms1 (Fig. 3E). These results demonstrate that membrane-associated Sms1 is mostly full-length and thus largely protected from proteolysis.

We noted that Sms1-sfGFP also localises to the nucleus (Fig. S3B), in contrast to the nuclear exclusion of Ste4, suggesting that a pool of Sms1 transits through the nucleus. To determine the functional importance of the different pools of Sms1 during mating, we engineered a system to retain Sms1 in the nucleus. Co-expression of GFP-binding protein (GBP)[91] tagged with a nuclear localisation sequence (NLS) retained Sms1-sfGFP in the nucleus, decreasing protein stability and drastically hindering mating efficiency (Fig. 3F–H). We conclude that nuclear export of Sms1 is necessary for its function. This is consistent with a functional role at the plasma membrane, where the protein is largely protected from proteolysis.

## The arrestin domain of Sms1 interacts with the plasma membrane, where it is stabilised by the Gα

Sms1 is predicted to be highly disordered and its only folded domain is a highly unusual arrestin fold (Fig. S4A, B). Whereas canonical arrestin proteins require the interaction between their two arrestin lobes to prevent untimely activation[92], Sms1 is predicted to have a single arrestin C-domain. However, this arrestin domain is remarkably similar in structure to the C-domain of the well-characterised classical arrestin proteins such as mammalian β-arrestin 2 (Fig. 4A). Interestingly, the Sms1 arrestin domain expressed in vegetative cells localised with the plasma membrane marker LactC2 (Fig. 4B)[93]. Based on the recent identification of hydrophobic residues in the C-edge of β-arrestins-2 that function together with positively charged PIP-binding residues for insertion in the lipid bilayer[94], we mutated the corresponding amino acids in Sms1 arrestin domain to create a lipid-binding deficient mutant (*sms1^LBM^*) (Fig. 4A), which failed to localise at the cell surface (Fig. 4B). Sms1 arrestin domain localisation to the plasma membrane was also abrogated in *its3-1*, a temperature sensitive mutant of the enzyme converting PI4P into PI(4,5)P2[95], indicating this is the major phosphoinositide species required for Sms1 membrane interaction. The lipid binding affinity of the arrestin domain is thus conserved in Sms1 and this property is essential for Sms1 function as shown by the sterility of a *sms1^LBM^* allele lacking the lipid-binding residues (Fig. 4D).

The targeting of Sms1 arrestin domain to the entire plasma membrane of mitotic cells raised the question of its restriction to the polarity patches and fusion site. A possible explanation would be an indirect recruitment by Ras1 because this GTPase binds the MAP3K^Byr2 54,96, localises to the polarity patches[67] and was proposed to recruit the MAP3K^Byr2 to the membrane[97]. However, upon deletion of Ras1 or its activator GEF Ste6, which strongly impair mating[98–100], Sms1 nevertheless polarised at the plasma membrane, forming unstable patches oriented towards WT partner cells (Fig. 4E). Thus, Sms1

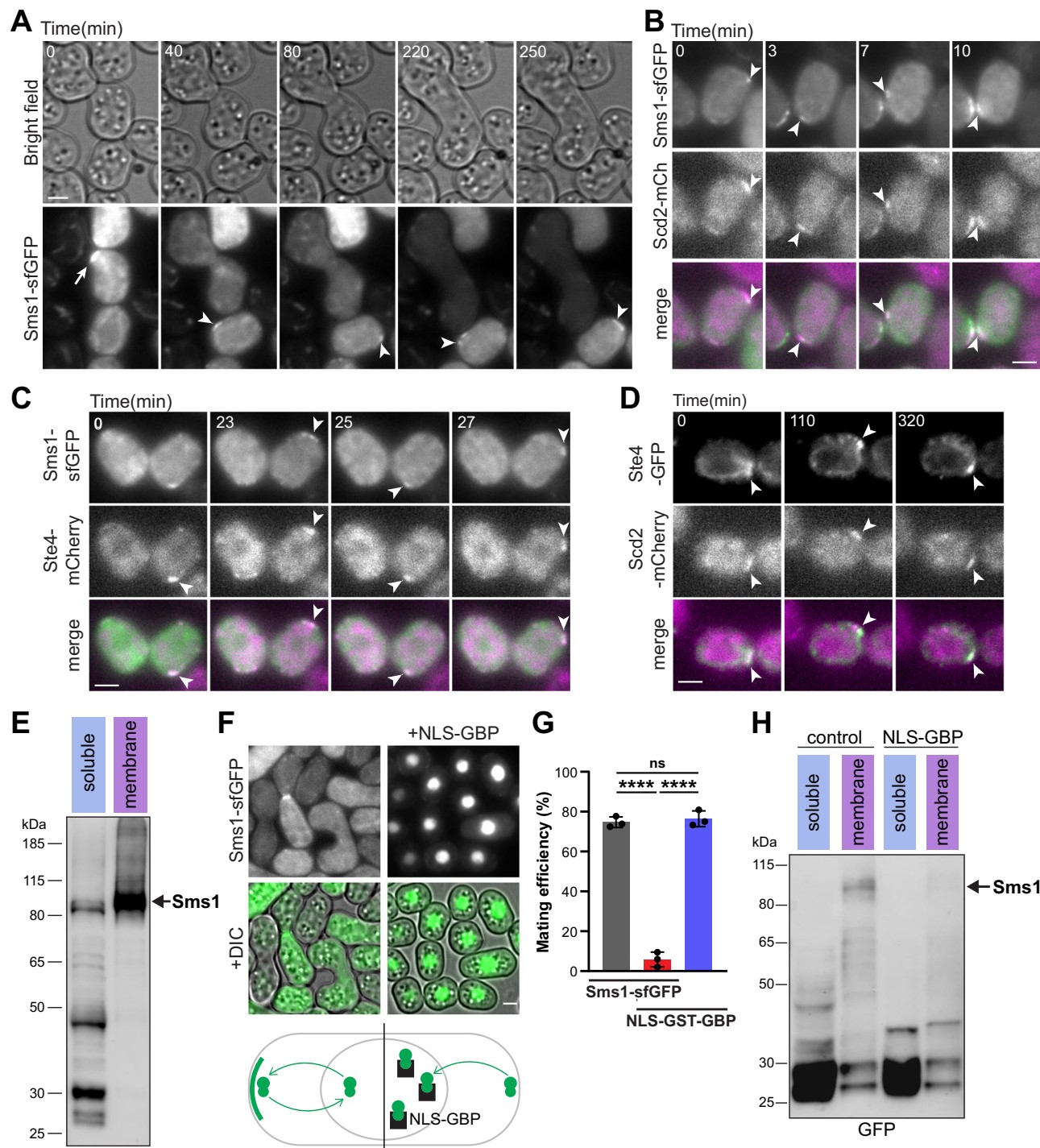

**Fig. 3 | Sms1 localises to membrane-associated polarity patches and the fusion site. A** Mating of *h- sms1-sfGFP*. Arrowheads and arrow indicate localisation of Sms1 in polarity patches and fusion site, respectively. Colocalisation of **B** Sms1-sfGFP and Scd2-mCherry, **C** Sms1-sfGFP and Ste4-mCherry and **D** Ste4-GFP and Scd2-mCherry in polarity patches (arrowheads) in *h90* mating cells. **E** Endogenous Sms1-sfGFP in membrane and soluble fractions from *h90* cells grown in MSL-N. Arrow indicates the size of full-length Sms1-sfGFP. **F** Localisation of Sms1-sfGFP in *h90* cells grown in MSL-N with or without expression of *P.tdh1-NLS-GST-GBP* to retain it in the nucleus, as shown on bottom scheme. Brightness and contrast in the presence of NLS-GBP are reduced to prevent signal saturation. **G** Mating efficiency of cells as in (**F**). $n \geq 500$ cells for three independent experiments with error bars as s.d. ****$P < 0.0001$; ns, not significant ($P = 0.8271$), one-way ANOVA with Tukey test. **H** Endogenous Sms1-sfGFP in membrane and soluble fractions from *h90* cells expressing or not *P.tdh1-NLS-GST-GBP* grown in MSL-N. Scale bars: 2 μm.

polarises to the plasma membrane upon pheromone signalling activation but independently of Ras1 activity.

As an alternative hypothesis, we investigated whether Sms1 might be recruited by the receptor-coupled Gα^Gpa1. When expressed in mating cells, Sms1 arrestin domain, but no other Sms1 fragment, was not only localised to the plasma membrane but was enriched at the fusion site, suggesting that the arrestin domain contains further localisation information (Figs. 4F and S4C). Co-immunoprecipitation showed that Sms1 interacts with Gpa1 in the membrane fraction and that this interaction occurs through the arrestin domain (Fig. 4G). Sms1

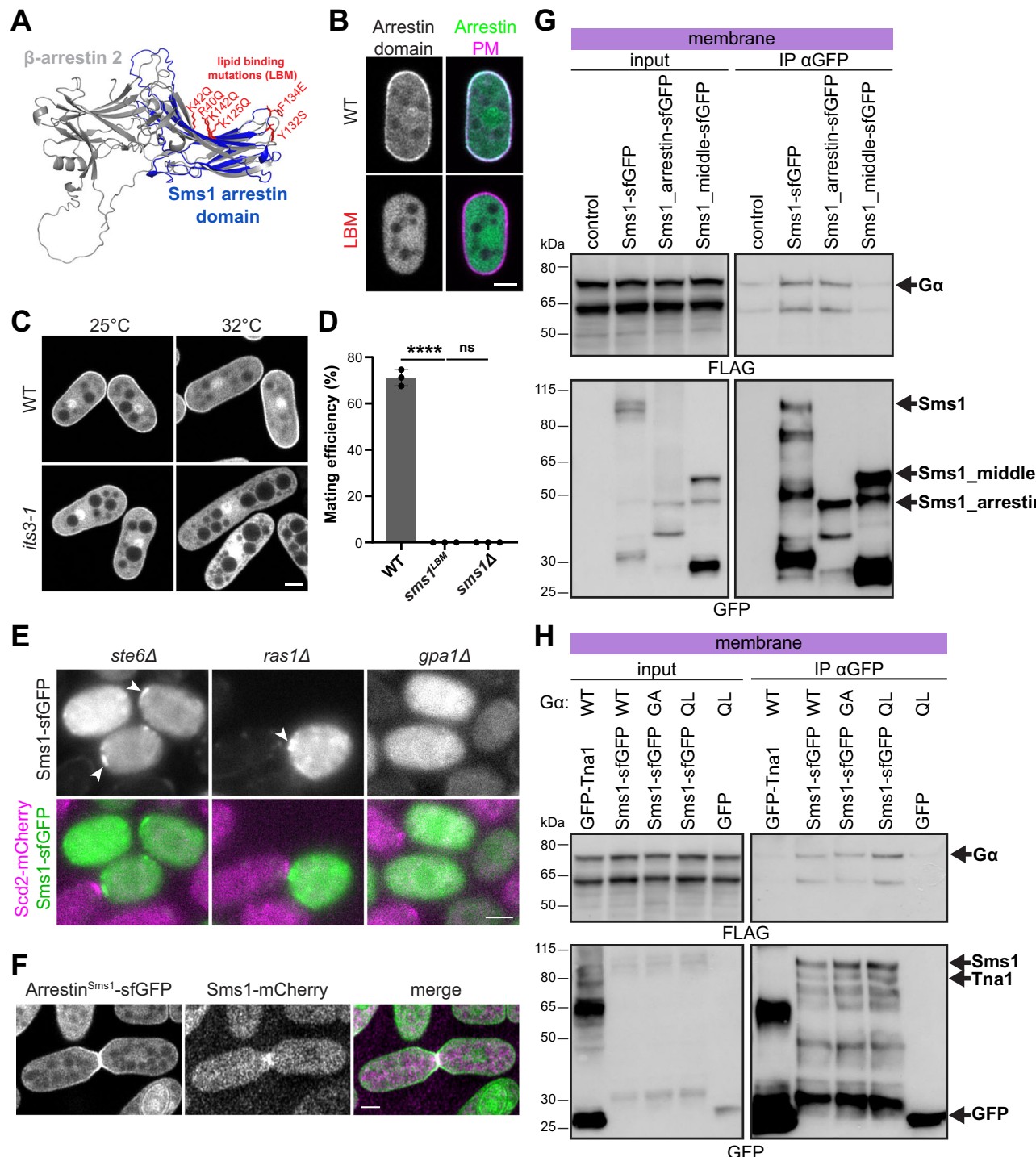

**Fig. 4 | The arrestin domain of Sms1 interacts with the plasma membrane, where it is stabilised by Gα. A** Sequence-independent structural alignment of Sms1 arrestin domain (blue) and human β-arrestin 2 (ARRB2; grey) AlphaFold2 predictions using PyMol-cealign with a root mean square deviation (RMSD) of 3.305 Å. Basic and hydrophobic residues mutated in the LBM mutant are highlighted in red. **B** Colocalisation of sfGFP-tagged Sms1 arrestin domain (WT or with LBM mutation) with the plasma membrane (PM) marker mCherry-LactC2 expressed in vegetative cells. **C** Localisation of Sms1 arrestin domain expressed in WT or *its3-1* vegetative cells. **D** Mating efficiency of *h90* cells with indicated *sms1* alleles at endogenous locus (*n* ≥ 500 cells for three independent experiments) with error bars as s.d. ****$P < 0.0001$; ns, not significant ($P > 0.05$), one-way ANOVA with Tukey test. **E** Sms1-sfGFP localisation in *h- ste6Δ, ras1Δ* and *gpa1Δ* cells mated with WT *h+*

Scd2-mCherry. Arrowheads indicate the localisation of Sms1 in polarity patches. Brightness and contrast of GFP in *ste6Δ* and in *ras1Δ* are reduced to prevent signal saturation. **F** Localisation of Sms1 arrestin domain expressed under *nmt41* promoter and native full-length Sms1-mCherry in mating cells. Gα co-precipitates with Sms1-sfGFP. Membrane fractions from vegetative cells expressing Gpa1-3xFLAG-mCherry and Sms1-sfGFP under the *nmt41* promoter were immunoprecipitated with anti-GFP beads and immunoblotted for FLAG or GFP. In (**G**), Sms1 full-length, arrestin domain and middle fragment (aa 158-397) were used. In (**H**), Gpa1 was either WT, GDP-locked (G242A; GA) or GTP-locked (Q244L; QL). Quantification is shown in Fig. S4D. No GFP construct (**G**), GFP and the unrelated transmembrane protein GFP-Tna1 (**H**) were used as negative controls. Scale bars: 2 µm.

similarly interacted with active GTP-locked Gpa1$^{Q244L}$ and inactive GDP-locked Gpa1$^{G242A}$ (ref. 77) (Figs. 4H and S4D). In agreement with Gpa1 recruiting Sms1, Sms1 polarisation was abolished in the absence of the Gα$^{Gpa1}$ (Fig. 4E). Together, these results demonstrate that Sms1 is targeted by its arrestin domain to the plasma membrane, where it recognises the Gα, thus restricting Sms1 to pheromone signalling domains.

## Sms1 is the MAPK scaffold of the pheromone signalling pathway

The interactions of Sms1 with phospholipids, Gα$^{Gpa1}$and Ste4 and their essential role in sexual reproduction indicate that Sms1 functions at the top of the pheromone-MAPK signalling pathway. Given the function of Sms1 in transducing the MAPK signal, we also tested its possible interaction with the MAP2K$^{Byr1}$ and MAPK$^{Spk1}$. To avoid expression of other mating signalling components, we overexpressed tagged proteins in vegetative cells and further deleted the transcription factor Ste11 to prevent sexual differentiation by constitutively active MAP2K$^{Byr1}$, in which the conserved MAP2K dual phosphorylation sites are mutated to aspartic acid (Byr1$^{DD}$, ref. 101). Co-immunoprecipitation experiments demonstrated that Sms1 binds the MAP2K$^{Byr1}$ (Fig. S5A). We also detected a strong interaction of Sms1 with active Byr1$^{DD}$ in the membrane fraction (Fig. 5A). Similarly, we found that Spk1 co-immunoprecipitates with Sms1-GFP but not GFP alone (Fig. 5B). Thus, Sms1 associates with both MAP2K and MAPK.

To probe for the complex formed during mating, we used the phospho-specific pERK antibody. Endogenous FLAG-tagged Sms1 co-immunoprecipitated the endogenous, active phopho-MAPK$^{Spk1}$ in the membrane fraction (Fig. 5C). Conversely, pERK beads pulled down endogenous Sms1—remarkably only the full-length protein—from the membrane fraction (Fig. 5D). Thus, the active MAPK$^{Spk1}$ specifically interacts with full-length Sms1, further supporting the stabilisation of the active Sms1 complex at the plasma membrane to ensure local MAPK activation.

Together, these data demonstrate that Sms1 is a novel MAPK scaffold, which promotes pheromone signalling by associating with every single component of the signalling cascade, including the Gα$^{Gpa1}$, the MAP3K$^{Byr2}$ through its adaptor Ste4, the active MAP2K$^{Byr1}$ and MAPK$^{Spk1}$. We thus renamed the protein as Scaffold for MAPK Signalling 1 (Sms1), which also avoids confusion with *S. cerevisiae* MAP2K. The scaffold is stabilised at the plasma membrane in association with the active kinase cascade.

To probe whether Sms1 is solely required for MAPK activation, we used *byr1$^{DD}$* strains, in which the native *byr1* gene encodes a constitutively active MAP2K$^{Byr1}$. In these conditions, we expected the MAPK to remain constitutively activated by Byr1$^{DD}$ irrespective of the presence of the scaffold. Indeed, loss of Sms1 did not significantly affect pSpk1 level (Figs. 5E and S5B) and only moderately decreased MAPK-dependent transcriptional response measured by the dPSTR assay (Fig. 5F). Phenotypically, *byr1$^{DD}$* at the endogenous locus causes a *fus* (fusion-defective) phenotype characterised by constitutive growth of mating projections (known as shmooing; Fig. 5G)[69,101]. These cells also show an accumulation of the cell wall. In *byr1$^{DD}$* cells, shmooing was abrogated by deletion of downstream signalling components, such as the MAPK$^{Spk1}$, but not of upstream ones, such as *ste4* (Fig. 5G). Unexpectedly, in the absence of Sms1, *byr1$^{DD}$* cells showed a total loss of the shmooing phenotype (Fig. 5G). This indicates that, in addition to acting as a scaffold that promotes MAPK signal transduction, Sms1 is also required for local MAPK output to induce the formation of the mating projection.

## Negative feedback by Spk1 removes Sms1 from the plasma membrane to prevent untimely mating attempts

The presence of high-molecular-weight bands suggested that Sms1 is regulated by phosphorylation (Fig. 3E). Indeed, these high-molecular-weight bands were largely abrogated by phosphatase treatment (Fig. 6A). Sms1 phosphorylation was further supported by its interaction in vegetative cells with the phospho-binding protein 14-3-3, Rad24 (Fig. S6A). To test if Sms1 might be regulated by proline-directed kinases such as the MAPK, we mutated to alanine all Sms1 serine and threonine residues followed by a proline (Sms1$^{22A}$) (Fig. 6A). High-molecular weight bands were not observed for Sms1$^{22A}$ and phosphatase treatment did not change its migration pattern, demonstrating that Sms1 expressed in vegetative cells undergoes proline-directed phosphorylation (Fig. 6A).

To probe the functional role of Sms1 phosphorylation, we expressed Sms1$^{22A}$-sfGFP as the sole copy under the endogenous promoter, similarly to Sms1$^{WT}$-sfGFP. While Sms1 formed patches at the membrane in mating cells (Fig. 3), it remained mostly cytosolic upon its starvation-induced expression in cells not exposed to pheromones (Fig. 6B). By contrast, we found that, upon starvation-induced expression, Sms1$^{22A}$-sfGFP was strongly enriched in membrane-localised patches (Figs. 6B and S6B). Remarkably, deletion of *MAPK$^{Spk1}$* was sufficient to induce a similar, though less extensive, increase in plasma membrane accumulation of Sms1$^{WT}$ (Figs. 6B and S6B). We conclude that MAPK$^{Spk1}$ likely phosphorylates Sms1 and prevents its accumulation at the plasma membrane.

Sms1$^{22A}$ exhibited gain-of-function phenotypes at all stages of the mating process. First, we focused on the response to the synthetic pheromone of a single mating partner. For these assays, we used M cells lacking the P-factor protease Sxa2, preventing degradation of the synthetic pheromone. In response to 10 nM P-factor, these cells exhibit dynamic polarity patches, to which Sms1-sfGFP localises (Fig. 6C)[61,62]. By contrast, *sms1$^{22A}$* cells formed stable, higher-intensity patches that were largely constrained to cell poles, often simultaneously at both cell poles (Figs. 6C and S6C; Supplementary Movie 1). These stable patches induced polarised growth and *sms1$^{22A}$* cells formed shmoos of increasing length at all P-factor dosages tested, even when WT cells did not (Fig. 6D). This hypersensitivity of *sms1$^{22A}$* cells to pheromone indicates that Sms1 phosphorylation normally limits the pheromone response.

Second, we examined mating initiation. Mating is induced by nitrogen starvation in fission yeast, and the presence of glutamate as a nitrogen source largely suppresses mating in WT cells[102,103]. By contrast, *sms1$^{22A}$* mutant cells presented hyperactivation of Spk1 during growth in glutamate-containing medium, leading to high mating in these conditions (Fig. 6E, F). This indicates that the very low levels of Sms1 during mitotic growth are normally phosphorylated, consistent with its association with 14-3-3 proteins and that this prevents mating activation.

Finally, we looked at mating termination in zygotes. In mating assays *sms1$^{22A}$* cells formed pairs and fused efficiently, but exhibited a striking phenotype upon cell-cell fusion. In contrast to the rapid signal disappearance of Sms1-sfGFP in zygotes, Sms1$^{22A}$-sfGFP remained polarised at the plasma membrane, leading to growth of the zygote towards other mating partners (Fig. 6G and Supplementary Movie 2). Thus, phospho-regulation of Sms1 is required to switch off signalling and to prevent inappropriate mating responses in zygotes.

Conversely, a phospho-mimetic allele, *sms1$^{22E}$*, abolished plasma membrane recruitment and led to a sterility phenotype (Fig. 6G). Sms1$^{22E}$ also blocked the *fus* phenotype of constitutively active MAP2K *byr1$^{DD}$* cells (Fig. S6D), further supporting a phosphorylation-dependent inhibition of Sms1. While we did not systematically map which of the 22 sites causes the described phenotypes, we found that the four phosphorylation sites in the arrestin domain play a key role, as Sms1$^{4A}$ showed localisation to the membrane of the zygote, while Sms1$^{4E}$ was sterile (Fig. 6A, H). We note that other phosphorylation sites also contribute to Sms1 function, as *sms1$^{4A}$* was less potent than *sms1$^{22A}$* in inducing persistent growth in zygotes.

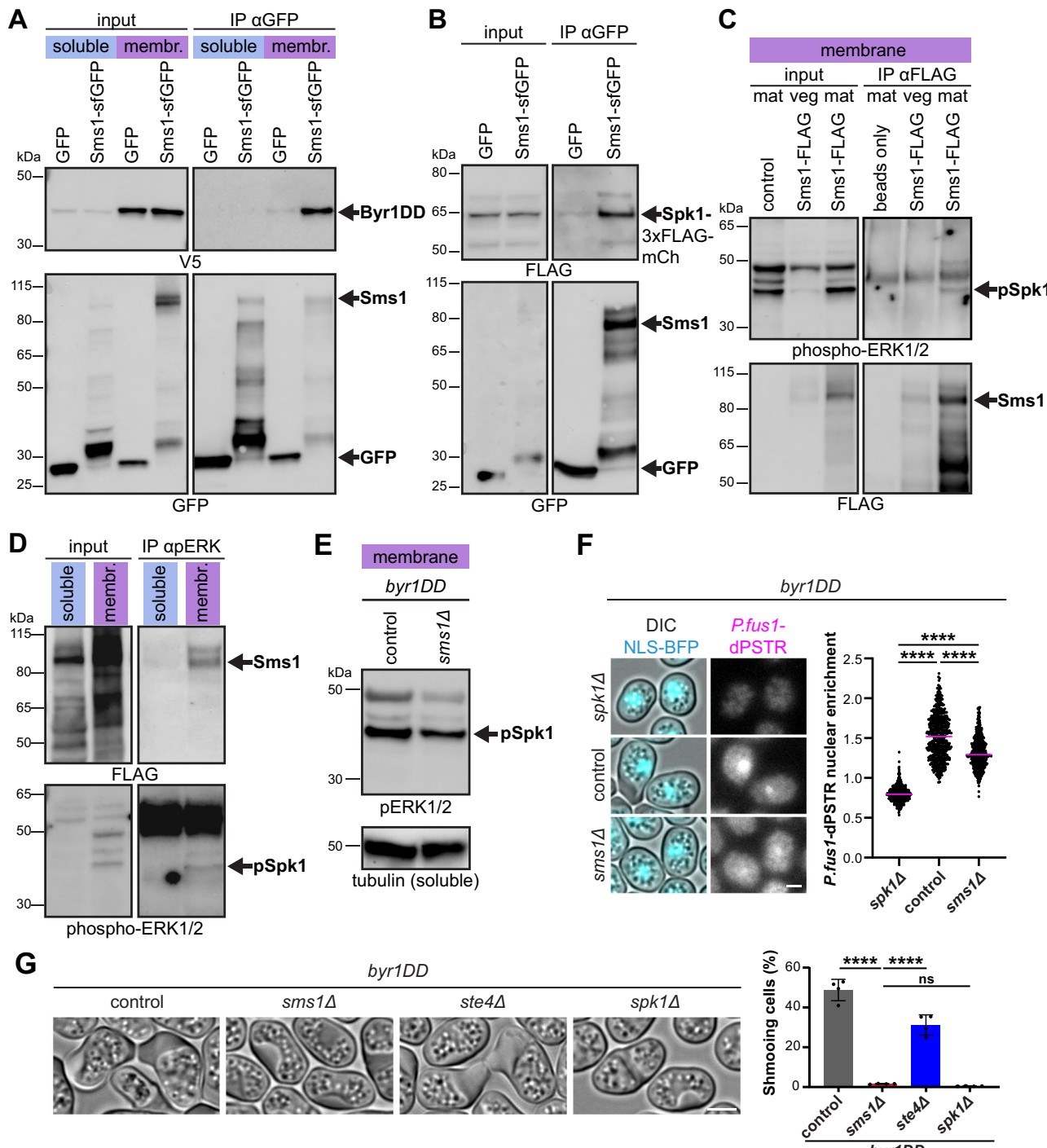

**Fig. 5 | Sms1 is the MAPK scaffold of the pheromone signalling pathway. A** V5-Byr1[DD] co-precipitates with Sms1-sfGFP. Membrane and soluble fractions from vegetative *ste11Δ* cells expressing V5-Byr1[DD] and Sms1-sfGFP under the *nmt41* promoter were immunoprecipitated with anti-GFP beads and immunoblotted for V5 or GFP. GFP serves as negative control. **B** Spk1-3xFLAG-mCherry co-precipitates with Sms1-sfGFP. Whole cell extracts from vegetative cells expressing Spk1-3xFLAG-mCherry and Sms1-sfGFP or GFP under *nmt41* promoter were immunoblotted for FLAG and GFP. **C** Phospho-Spk1 co-precipitates with endogenous Sms1-3xFLAG-mCherry. Membrane fractions from endogenous Sms1-3xFLAG-mCherry *h90* cells grown in MSL + N(veg) or MSL-N(mat) were immunoprecipitated with anti-FLAG beads and immunoblotted for phosphorylated Spk1 (anti-pERK1/2) and FLAG. Note that part of the input blot is also shown in Fig. 1B. **D** Endogenous Sms1-3xFLAG-

mCherry co-precipitates with phospho-Spk1. Membrane and soluble fractions from *h90* cells in mating conditions (MSL-N) were immunoprecipitated with anti-pERK1/2 beads and immunoblotted for FLAG and pERK1/2.**E** Phospho-Spk1 (arrow) in membrane fractions from *h90 byr1[DD]* and *h90 byr1[DD] sms1Δ* cells in MSL-N. Quantification is shown in Fig. S5B. **F** Quantification of nuclear enrichment of the dPSTR reporter in *h90 byr1[DD]* with *spk1Δ*, *sms1Δ* or otherwise WT (control) after 24 h on MSL-N with median (magenta), $n \geq 500$ cells for two independent experiments. ****$P < 0.0001$, Kruskal-Wallis with Dunn's test. **G** DIC images and shmooing efficiency of *h90 byr1[DD]* with *sms1Δ*, *ste4Δ*, *spk1Δ* or otherwise WT (control) in MSL-N ($n \geq 500$ cells for four independent experiments). Cell wall accumulation was present in all conditions. Error bars represent s.d. ****$P < 0.0001$; ns, not significant ($P = 0.9731$), one-way ANOVA with Tukey test. Scale bars: 2 µm.

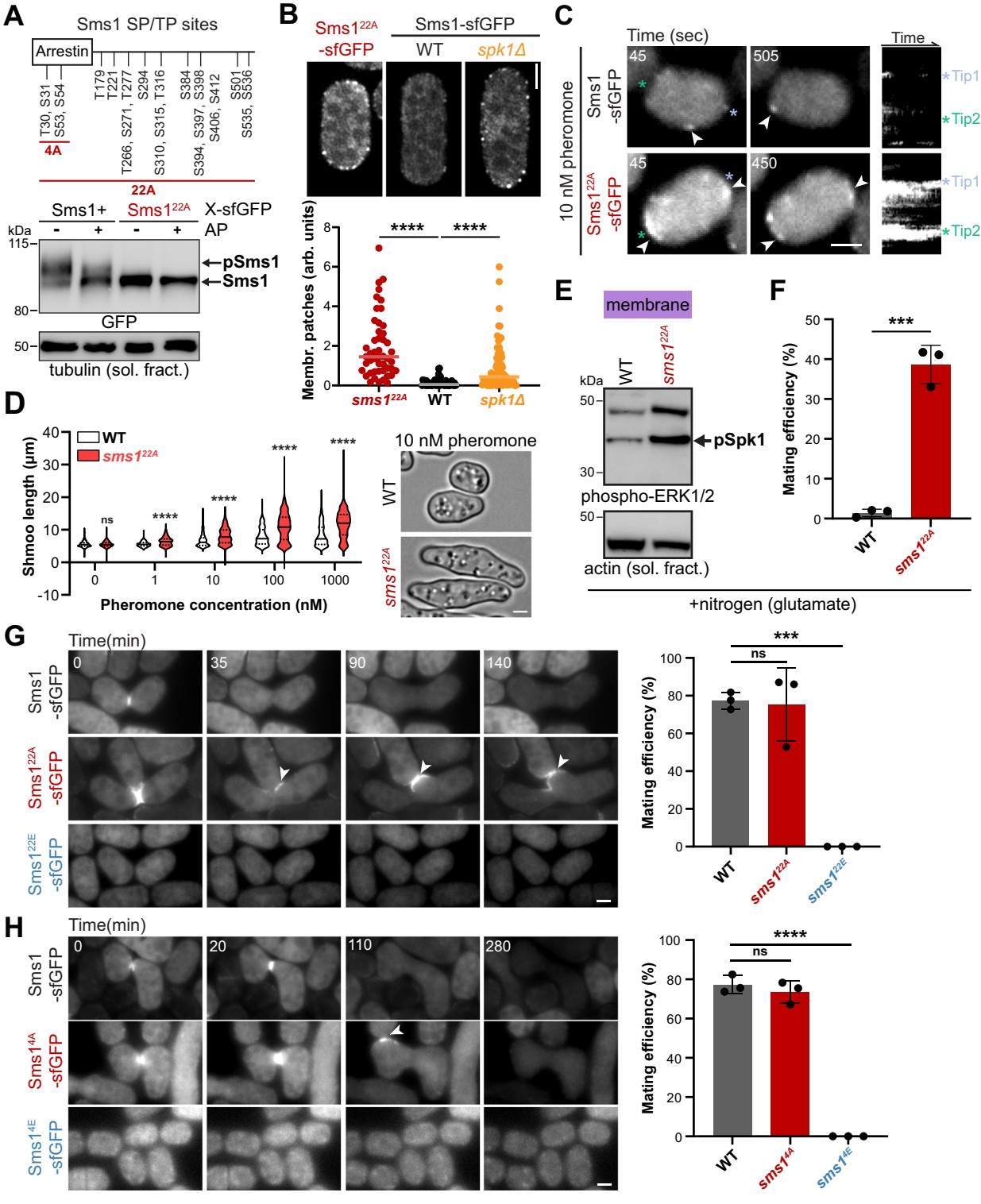

Together, our results show that the Sms1 scaffold undergoes a negative feedback loop where its associated MAPK counteracts its excessive surface accumulation, preventing untimely mating attempts.

## Discussion

In this paper, we identify Sms1 as a structurally novel MAPK scaffold essential for fission yeast mating. Sms1 exhibits key features of a scaffold protein: (i) it interacts with all three MAP kinases, (ii) it localises to the plasma membrane, where it assembles the MAPK cascade for local signalling, and (iii) it is negatively regulated by

phosphorylation to promote membrane detachment and signalling arrest. Our data support the following model of the regulatory steps of the MAPK scaffold Sms1 (Fig. 7A). Transient interactions of the Sms1 hemi-arrestin domain with phospholipids at the plasma membrane allow it to survey the cell surface. Upon pheromone perception in proximity of a mating partner, Gα stabilises Sms1 at the membrane, forming a polarity patch. Sms1 recruits the MAP3K adaptor, the MAP2K and the MAPK, assembling the MAPK cascade. On Sms1, spatial proximity of the kinases facilitates their sequential activation, transducing the signal to trigger downstream global signalling and bringing

**Fig. 6 | Negative feedback by phosphorylation removes Sms1 from the plasma membrane to prevent untimely mating attempts. A** Membrane fractions from lysates of Sms1-sfGFP and Sms1^22A-sfGFP expressed under *nmt41* promoter in vegetative *ste11Δ* cells before and after alkaline phosphatase (AP) treatment. A schematic of Sms1 with all SP and TP sites, indicating those mutated in Sms1^22A/E and Sms1^4A/E, is shown at the top. **B** Localisation of native Sms1^22A-sfGFP or Sms1-sfGFP in *h+* starved WT and *spk1Δ* cells. Quantification of the intensity of membrane patches is shown at the bottom (*n* > 40 cells for two independent experiments). ****$P$ < 0.0001. Quantification of the number of patches is shown in Fig. S6B. **C** Time points and kymographs from time-lapse imaging of Sms1-sfGFP and Sms1^22A-sfGFP in *h- sxa2Δ* cells treated with 10 nM P-factor for 60 min. Arrowheads indicate polarity patches. Asterisks indicate cell poles. Contrasts are identical in the kymographs of both strains to illustrate the different amounts of protein at the membrane, but reduced for Sms1^22A-sfGFP time points to prevent signal saturation.

Additional examples are shown in Fig. S6C. **D** Quantification of shmoo length in *h-sxa2Δ sms1+* or *sms1^22A* cells upon treatment with 0-1000 nM P-factor for 24 h (*n* > 350 cells). ****$P$ < 0.0001; ns, not significant (*P* > 0.05) with the central line as the median and dotted lines as quartiles. **E** Phospho-Spk1 in membrane fractions from *h90 sms1+* or *sms1^22A* grown in MSL+Glutamate. **F** Mating efficiency of strains as in (E) of *n* ≥ 500 cells for three independent experiments with error bars as s.d. ***$P$ = 0.0002. Time-lapse imaging of (**G**) *sms1^22A* and *sms1^22E* and **H** *sms1^4A* and *sms1^4E h90* mutant cells during mating. WT and mutant forms of Sms1 are tagged with sfGFP at the endogenous locus. Arrowheads point to ectopic polarisation in the zygote, leading to polarised growth towards other mating partners. Brightness and contrast are reduced for Sms1^22A-sfGFP to prevent signal saturation. The strains' mating efficiency is shown on the right (*n* ≥ 500 cells for three independent experiments) with error bars as s.d. ***$P$ = 0.0004, ****$P$ < 0.0001; ns, not significant (*P* > 0.05). Scale bars: 2 μm.

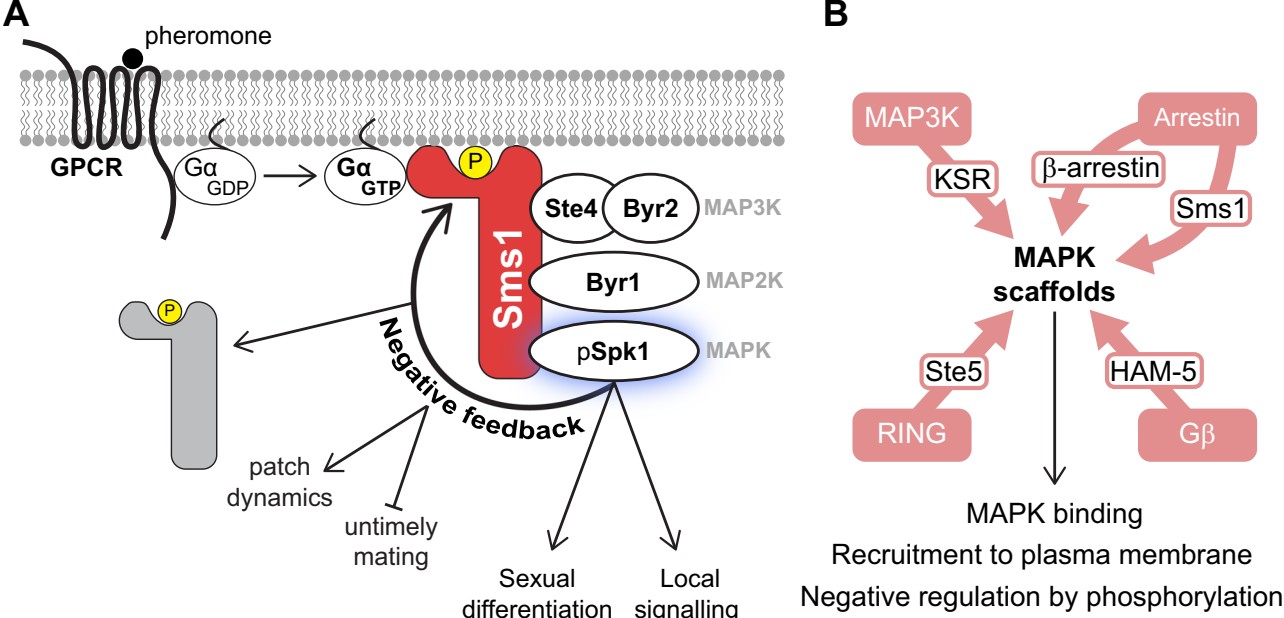

**Fig. 7 | Model of Sms1 scaffold function. A** Sms1 is recruited to sites of pheromone receptor activation through PI(4,5)P2 and Gα binding, where it assembles the MAPK cascade. This promotes local activation of the MAPK^Spk1, which induces sexual differentiation and local signalling for pheromone-directed growth. MAPK^Spk1 also phosphorylates Sms1, promoting its detachment from the membrane and signal arrest. This negative feedback is critical to reduce pheromone sensitivity and promote patch dynamics during mating and to prevent untimely mating of vegetative cells and zygotes. **B** Although sharing essential roles in MAPK binding and membrane recruitment and negatively regulated by phosphorylation, MAPK scaffolds across phyla have diverse evolutionary origins.

the activated MAPK in proximity to local substrates to promote morphogenesis. To limit signalling, the MAPK in turn phosphorylates Sms1, triggering its release from the plasma membrane. This negative feedback loop promotes exploratory patch dynamics and restricts signalling to prevent untimely mating behaviours.

### Roles of the Sms1 scaffold

Because Sms1 binds Gα and all MAPKs at the plasma membrane, it constitutes a key adaptor between pheromone-receptor engagement and downstream signal transmission, thus ensuring the MAPK cascade is activated at sites of pheromone perception. Localised membrane-binding occurs upon coincidence detection of phospholipids and the Gα, which is conceptually similar to Ste5 binding phospholipids and free Gβγ[17,29], although whether and how Sms1 specifically engages with the receptor-activated Gα remains to be established. Membrane recruitment is essential for Sms1 function, in the absence of which there is no MAPK activation. Mechanistically, a key step to activate the MAP3K^Byr2 is its binding by Ras1-GTP, similar to Raf activation by Ras GTPase in mammalian cells[55,104–106]. A second MAP3K^Byr2 activation event may involve phosphorylation by p21-activated kinase[55], similar to MAP3K activation in *S. cerevisiae*[107]. Both reactions take place at the

plasma membrane, to which Sms1 recruits MAP3K^Byr2 via Ste4. This is consistent with previous genetic epistasis showing that Ste4 and Ras1 form two additive inputs to MAP3K^Byr2 activation[54,55]. Thus, one role of Sms1 is as an adaptor that recruits and concentrates the MAPK cascade to sites of receptor engagement in the vicinity of Ras1 at the membrane.

This raises the question whether Sms1 'simply' acts as a membrane adaptor to initiate signalling or whether it is also important to transduce the signal through the consecutive kinases. We have shown that, when the MAP2K is constitutively activated, Sms1 only plays a minor role in the activation of the MAPK^Spk1. While this result suggests that Sms1 facilitates MAPK^Spk1 phosphorylation by the MAP2K, perhaps by bringing them in proximity to favour their direct interaction[108], it speaks against a major role of Sms1 in allosteric MAPK activation. This is different from the allosteric role of the budding yeast scaffold Ste5, which is required for phosphorylation of MAPK Fus3 (but not the second MAPK Kss1) even in the presence of activated MAP2K[20,23,109]. Ste5 evolution and allostery were proposed to coincide with MAPK duplication and re-use of MAPK in distinct cascades in budding yeast[45,110]. The mechanistic difference we observe in Sms1 is in line with the observations that there is no MAPK duplication in fission yeast, nor

re-use of the same MAPK in distinct cascades, simplifying the specificity of signal transmission. Whether Sms1 is necessary for activation of the MAP2K$^{Byr1}$ through allosteric effects, as observed for KSR[24], awaits future exploration.

One interesting observation is that, even though Sms1 is not essential for MAPK$^{Spk1}$ activation by the hyperactive MAP2K Byr1$^{DD}$ nor for global transcriptional output, the scaffold is essential for the local polarised response. Thus, Sms1 has a critical function in promoting local signalling. This function could arise from positioning the cascade at the right place, thus ensuring local activation of relevant substrates at the cell cortex to drive polarised growth. Sms1 may also directly promote substrate specificity by recruiting substrates to the MAPK. One question this raises is whether Sms1 may be instructive in linking receptor activation to the polarity machinery. In principle, given the known interactions of Sms1-Ste4- MAP3K$^{Byr2}$-Ras1[54,55], Sms1 could recruit Ras1 via Ste4- MAP3K$^{Byr2}$, but this hypothesis is inconsistent with the observation that byr1$^{DD}$ shmoo formation is not impaired by ste4 deletion. Furthermore, Sms1 forms restricted patches towards mating partners even in ras1Δ cells, indicating spatial signal interpretation independently of Ras1 GTPase. In budding yeast, two distinct scaffolds with similar domain composition, Ste5 and Far1, link the Gβγ to the MAPK cascade and the Cdc42 polarity module, respectively. However, neither exists in S. pombe[45]. Future experiments should establish the roles of the Sms1 scaffold in the spatial interpretation of the pheromone signal.

## A negative feedback loop on Sms1 prevents MAPK hyperactivation

Negative regulation of plasma membrane association by phosphorylation is a recurring feature in MAPK scaffolds[7]. In keeping with this theme, phosphorylation of proline-directed serine and threonine residues strongly inhibits Sms1 membrane association. Reflecting the major role of the hemi-arrestin domain in binding the membrane, this domain contains up to four critical phosphorylation sites. Future work should evaluate how phosphorylation interferes with membrane binding, as these sites are not in immediate proximity to the phospholipid-binding basic residues on the concave side of the arrestin domain. Because Sms1 membrane localisation is antagonised by the MAPK$^{Spk1}$, a likely scenario is that MAPK$^{Spk1}$ targets (some of) these sites to promote Sms1 membrane detachment. Thus, a negative feedback loop—where Sms1 promotes the activation of MAPK$^{Spk1}$ at the cortex, which in turn induces Sms1 membrane detachment— regulates MAPK output.

Negative regulation of the MAPK scaffold is critical to prevent activation by spurious environmental noise and to prevent response saturation[111]. Mating in fission yeast relies on the formation of a dynamic polarity patch that undergoes assembly-disassembly cycles at the cortex to sample potential mating partners[61,62]. We show that the Sms1-MAPK$^{Spk1}$ negative feedback contributes to patch instability by destabilising the patch. Indeed, in non-phosphorylatable sms1$^{22A}$ cells, patches are long-lived and more intense, resulting in pheromone hypersensitivity. This negative feedback likely combines with other negative regulations of patch components, including inactivation of Ras1 GTPase by its GAP Gap1[67] and of Cdc42 GTPase by its GAP Rga3[112]. One open question is how the patch stabilises despite negative feedback on Sms1 at the incipient fusion site. One possible scenario could be the pheromone-dependent recruitment of an antagonist phosphatase, similar to the stimulated recruitment of PP2A that stabilises KSR1 at the membrane[21].

Remarkably, Sms1 phosphorylation is also crucial to prevent untimely activation of the pathway in rich medium and to terminate signalling at the end of mating. Low-level Spk1 expression may help repress stochastic pathway activation during mitotic growth in nitrogen-rich medium. More likely, however, additional kinases active in these conditions contribute to Sms1 phosphorylation. Sms1 has

been shown to interact with the kinase Pat1[70], which represses mating and meiosis initiation in presence of nitrogen[113,114] and thus constitutes a possible candidate for Sms1 repression in rich medium. This would be similar to the regulation of Ste5, which is phosphorylated by both CDK1 and the MAPK Fus3 on partially overlapping sites to antagonise membrane binding[35,36,39,115]. Establishing how Sms1 is rapidly phosphorylated post-fusion to prevent zygote mating is a key open question. Beyond phosphorylation, Sms1 must also be degraded for meiosis to take place in the zygote, as Sms1 overexpression inhibits meiosis, likely through its interactions with Pat1 kinase and the master regulator of meiosis, Mei2[70]. In summary, Sms1 is both an essential component of the mating MAPK cascade and a central regulatory hub that ensures timely mating behaviours.

## Sms1 hemi-arrestin and IDR as key determinants of its scaffolding function

Sms1 presents a single arrestin domain, which can be considered a hemi-arrestin as opposed to the canonical two-domain arrestin fold of the arrestin family. This hemi-arrestin domain provides affinity for phosphoinositides at the membrane via the presence of a basic patch on the concave side, a characteristic conserved throughout the arrestin family (Fig. S6E). The Metazoa-specific β-arrestin subgroup emerged from the more ancient α-arrestins, which are present throughout eukaryotes and typically recruit ubiquitin ligases through a PY motif to promote the ubiquitination and subsequent degradation of receptors and other membrane proteins[44,116]. Our work defining Sms1 as a MAPK scaffold, the observation that it does not contain a PY motif and our inability to detect interaction with the pheromone receptor by co-immunoprecipitation or two-hybrid assay, indicate that Sms1 does not play a typical α-arrestin function. Because β-arrestins are only found in Metazoa, Sms1 and β-arrestins likely independently acquired MAPK interfaces to function as signalling scaffolds (Fig. 7B). The similarity may even extend further, as suggested by the presence on the convex side of Sms1 hemi-arrestin of a DEF MAPK-docking motif, which structurally closely aligns with the lariat loop of β-arrestin 1 recently identified as a MAP kinase interface (Fig. S6E)[19].

Sms1 scaffolding function also requires its extensive intrinsically disordered regions (IDRs) in the C-terminus, notably for interaction with the MAP3K adaptor Ste4. Whether Sms1 also directly binds the MAP3K has not been tested. IDRs may facilitate evolutionary convergence as they provide structural and conformational plasticity for the appearance of multivalent interactions with the kinases, for example, upon coupled binding. In addition, IDRs are particularly prone to be regulated by multisite phosphorylation, thus favouring feedback regulation by the kinases and leading to the formation of signalling switches[87].

## Convergent evolution of the MAPK scaffolds

Beyond the evolutionary re-use of the arrestin fold in a MAPK scaffold, Sms1 displays extensive functional and regulatory similarities with the RING-containing budding yeast Ste5 and the MAP3K-derived mammalian KSR1/2 proteins, with whom it shares no sequence or structural homology. This discovery further establishes evolutionary functional convergence as a principle for the emergence of MAPK scaffolds (Fig. 7B)[7,117]. We want to emphasise that the convergent evolution of Sms1, Ste5 and KSR1 does not only concern their affinity for the MAP kinases, but also specific features of these proteins, such as their translocation to the plasma membrane and their inhibition by phosphorylation. This similarity is particularly striking between Sms1 and Ste5, which both scaffold a MAPK module that transduces pheromone signalling, which elicits highly similar physiological processes in divergent yeasts of the ascomycete lineage. Yet another MAPK scaffold, HAM-5 with a Gβ-like fold, is proposed to link a functionally similar MAPK cascade underlying hyphal fusion in filamentous

ascomycetes[118,119]. Remarkably, although not evolutionarily conserved, each of these proteins has become indispensable for signalling through the MAPK cascade that they scaffold. This raises a fundamental conundrum about their evolution. Did ancestral cells contain scaffolds, which were lost and supplanted by new ones in extant species? Or did scaffolds simultaneously appear in various lineages? In either case, under what evolutionary pressures were these new, essential components co-opted?

# Methods

## AlphaFold predictions and structural alignments
Interaction between Sms1 C-terminus (aa 350-569) and Ste4 RA domain (aa 176-264) was predicted using AlphaFold-multimer on ColabFold with 20 recycles[84]. Sequence-independent alignments between the arrestin domain of Sms1 and β-arrestin 1 or 2 were performed by incremental combinatorial extension (CEAlign)[120] on PyMol (Schrödinger).

## Strains, media and growth conditions
Strains used in this study are listed in Table S2. Transformations of *S. pombe* and *S. cerevisiae* cells were done according to standard genetic manipulation methods. Cells were grown in MSL supplemented with appropriate amino acid and nitrogen source (ammonium (N) or glutamate (E)) for at least 18 h at 30 °C to reach optical density at 600 nm (OD600) between 0.5 and 1[102,121]. For mating experiments, cells grown in MSL + E were washed three times with MSL-N and resuspended in MSL-N at an OD600 of 1.6. For stimulation with P-factor, cells were pregrown in MSL + N and similarly washed with MSL-N. Heterothallic strains were grown separately and mixed at a 1:1 ratio just before MSL-N washes. To avoid flocculation in MSL + N, *byr1^{DD}*-expressing strains were grown in YE until being washed and resuspended in MSL-N. The temperature-sensitive *its3-1* cells were grown in MSL + N at 25 °C, mounted on MSL + N 2% agarose pad and incubated at 25 °C or 32 °C for 5 h before imaging.

## Yeast two-hybrid assay
All plasmids were generated using standard molecular biology techniques. Two-hybrid assays were performed by co-transformation of the pGAD and pGBD plasmids containing the *ste4*, *sms1* and *byr2* variants in the AH109 host strain (Clontech). Interactions were assessed by growth on selective SD medium lacking histidine and supplemented with 3-amino-1,2,4-triazole (3AT).

## Pil1 co-recruitment assay
All plasmids were generated using standard molecular biology techniques and stably integrated into the genome. Fusion constructs are expressed under the *nmt41* promoter and comprise the gene of interest fused in frame with the *pil1* ORF at the 3' end. Expression was induced by the growth of cells in MSL + N and the bottom z-section of cells is shown.

## Mating assays
Cells were grown for 36 h in MSL + E at 30 °C to reach an optical density at 600 nm (OD600) between 0.5 and 1. Cells were washed three times with MSL-N and resuspended in MSL-N at an OD600 of 1.6. Cells were mounted on an MSL-N 2% agarose pad before incubation at 30 °C for 24 h or at 25 °C for 48 h. Mating efficiency was measured as the number of mating pairs and zygotes multiplied by two and divided by the total number of cells, as previously described[121]. Mating efficiency in MSL + E (Fig. 6E, F) was performed similarly without MSL-N washes and imaging was performed directly after mounting the cells on an MSL + E 2% agarose pad. Shmooing efficiency was measured as the number of cells exhibiting polarised growth divided by the total number of cells. For mating and shmooing efficiencies, for each condition, cells were counted from microscopy images until reaching 500 cells and

including the remaining cells in the last image ($n \geq 500$ cells per condition). This quantification was repeated in at least three biological experiments. Mating and shmooing efficiencies are shown as bar graphs for the mean of the independent experiments (dots) and the error bars as the standard deviation.

## Microscopy
All images in Figs. 1, 3–6 and Fig. S1, S3–S6 (except Figs. 4B, C, F, 6B and S4C) were acquired on a DeltaVision platform (Applied Precision) composed of a customised inverted microscope (Olympus IX-71), a UPlan Aporchromat 100x/1.4 NA oil objective, a camera (Photometrics CoolSNAP HQ2) and a colour combined unit illuminator (Insight SSI 7, Social Science Insights). Images were acquired using softWoRx v4.1.2 software (Applied Precision). Softwares used for data acquisition are listed in Table S4.

Super resolution imaging in Figs. 2C, 4B, C, F, and 6B; S2B and S4C were performed using a Zeiss LSM 980 scanning confocal microscope fitted on an inverted Axio Observer 7 microscope with Airyscan2 detector optimised for a 63x/1.40 NA oil objective and 488 nm and 561 nm lasers. Imaging was set in super resolution mode with a maximum of 7 frames per scanning speed. Laser power was used at <2.5% with pixel time below 2.3 µs. Images were acquired by the ZEN v.3.3 Blue software (Zeiss).

## Image analysis
**dPSTR**. For quantification of dPSTR signal in Figs. 1H, 5F and S1C, three-channel images (DIC, mCherry and DAPI) were acquired on a DeltaVision microscope. The DIC channel was used to determine the cell contour, the mCherry channel labels the dPSTR signal and the DAPI channel allows visualisation of the nucleus by detecting NLS-mtagBFP2.

For experiments in Figs. 1H and S1C, heterothallic WT, *spk1Δ* or *sms1Δ* cells expressing the dPSTR construct (*P.pom1*: mCherry-SynZip1 *P.fus1*: NLS-SynZip2) together with NLS-mtagBFP2, were mixed with untagged heterothallic cells of opposite mating type. For the experiment in Fig. 5F homothallic *byr1^{DD}*, *byr1^{DD} spk1Δ* or *byr1^{DD} sms1Δ* cells expressing the dPSTR construct (*P.pom1*:mCherry-SynZip1 *P.fus1*:NLS-SynZip2) together with NLS-mtagBFP2 were used. Images were acquired 24 h after MSL-N washes.

Individual cell masks were semi-automatically generated from the DIC image using the Segment Anything Model (SAM)[122]. To identify nuclei within these cells, candidate regions were extracted from the BFP channel. Nuclear detection was then performed using an automated pipeline combining convolutional filtering, percentile-based thresholding and morphological post-processing, ensuring robust segmentation of nuclear areas.

The resulting nuclear mask was compared with the cell mask to define both nuclear and cytoplasmic compartments, with the cytoplasm defined as the cell mask excluding the nucleus. These masks were overlaid with the dPSTR fluorescence signal in the mCherry image, enabling quantitative extraction of mean fluorescence intensities. To normalise for background signal, the nucleus-to-cytoplasm ratio (N/C) was computed as follow: $(I_{nuc} - I_{bg}) / (I_{cyto} - I_{bg})$, where $I_{nuc}$, $I_{cyto}$ and $I_{bg}$ represent the mean fluorescence intensities of the nucleus, cytoplasm and background, respectively. The background signal was estimated as the signal excluding tagged cells.

**Colocalisation in the co-recruitment assay.** The Pearson correlation coefficient was used to quantify colocalisation in the Pil1 co-recruitment assay in Figs. 2C and S2B as previously described[85]. In brief, background and cytosolic signals were subtracted in each channel. For each condition, at least 10 individual cells were manually selected as regions of interest (ROIs) using the freehand selection tool. Pearson's *R* values (no threshold) for each ROI were obtained using Fiji's Coloc 2 plugin.

**Measure of fluorescence zygotic intensity of Sms1 and Ste4.** To quantify Sms1 and Ste4 fluorescence intensities in zygotes, movies were aligned using the Fiji MultiStackReg plugin[123] to minimise stage drift. In both channels, background subtraction was performed using a Rolling Ball algorithm with a radius equal to the width of a cell. An elliptic region of interest (ROI) was defined to measure cytoplasmic fluorescence in the zygote, in a region that does not overlap with the nuclei at any of the time points. Mean grey values of the ROIs were measured for each timeframe from the first frame after cell-cell fusion (defined as the loss of GFP signal at the fusion site). For each channel, fluorescence intensities were normalised by the average signal of the first frame after cell fusion.

**Fluorescence intensity of Sms1-sfGFP membrane patches.** Quantification of membrane patches of Sms1-sfGFP or Sms1[22A]-sfGFP in Fig. 6B was performed on images acquired on an LSM980 Zeiss confocal microscope with Airyscan2 with 16x averaging. For quantification of single cells, a 3-pixel wide segmented line was designed around the total cell contour to quantify the signal at the membrane and an additional 3-pixel wide segmented line was created to measure internal signal. Normalisation was performed by dividing the perimetral fluorescence intensity at each point by the average internal signal of each cell. Normalised intensity of the membrane patches was measured for at least 40 cells per strain. The area under the curve of the fluorescence intensity was determined for each strain with a baseline of 1.5 using GraphPad Prism. The membrane patches (total peak area per cell) were plotted in Fig. 6B and their frequency (number of peaks per cell) was plotted in Fig. S6B.

**Shmoo length.** Quantification of shmoo length in Fig. 6D was performed on binary cell masks from DIC images acquired on a DeltaVision microscope. The masks were first skeletonised using a previously described approach[124]. To increase robustness, secondary skeleton branches were removed, leaving only the main trunk. The endpoints of this trunk were then identified and the connecting path was extended until it intersected with the cell boundary at both tips, ensuring that the axis spanned the full cell length. Finally, spline interpolation was applied to the extended path to generate a smooth, continuous curve representing the major axis of the cell. Total cell length was defined as the length of this major axis. For each strain, the shmoo length for each condition was determined by subtracting the average cell length of untreated cells (0 nM P-factor).

**SDS-PAGE and Western blotting**
For experiments shown in Fig. S3C, D, RIPA buffer (50 mM Tris-HCl pH7.5, 150 mM NaCl, 1 mM EGTA, 1 mM MgCl2, 0.1% SDS, 0.4% sodium deoxycholate, 1% NP-40) was supplemented with protease inhibitors (protease inhibitor cocktail (Roche 11836145001), 0.7 μM Antipain, 0.08 μM Aprotinin, 0.8 μM Chymostatin, 1 μM Leupeptin, 0.7 μM Pepstatin A, 5 μM 1,10 Phenanthroline, 3.5 μM Benzamidine-HCl, 1 mM PMSF (Phenylmethylsulfonyl fluoride)). 100 ODs of cells were lysed in 500 μl of RIPA buffer by 8 cycles of 30 s of bead-beating at 4 °C with 30 s between cycles. Lysates were cleared by centrifuging at 13'000 × g for 20 min at 4 °C before transferring the supernatants into collection tubes.

For all other experiments, lysis buffer (LB, 50 mM Tris-HCl, pH 7.4, 200 mM NaCl, 1 mM EDTA) was prepared by supplementing it with the protease inhibitors described above (LBi). PhosSTOP (Roche, 4906837001) was also added in LBi for the detection of phospho-Spk1. 100 ODs of cells were lysed in 400 μl of LBi + 1% LMNG (Lauryl Maltose Neopentyl Glycol) + 0.1% CHS (Cholesteryl Hemisuccinate) by 4 cycles of 30 s of bead-beating at 4 °C with 2 min on ice between cycles. Cell lysates were eluted into collection tubes by centrifuging pierced tubes at 400 g for 20 min at 4 °C. Lysates were cleared by centrifuging at 13'000 × g for 20 min at 4 °C.

For all experiments, protein concentration was determined by Bradford assay. Equalised protein amounts were denatured at 65 °C for 15 min in 4x NuPAGE LDS sample buffer containing 5% β-mercaptoethanol. Samples were run in 4–12% and 4–20% Bis-Tris SurePAGE gradient gels (Genscript) and MOPS running buffer (Genscript M00138). Proteins were transferred onto a 0.45 μm nitrocellulose membrane (Cytiva). 5% milk in TBST (0.05% Tween-20) was used for blocking and antibody incubation and TBST was used for washing. The list of the antibodies used in this study is available in Table S3. Phospho-ERK1/2 (Thr202/Tyr204) antibody cross-reacting with phospho-MAPK[Spk173] (Fig. S1B) was used to detect its phosphorylated conserved TEY motif.

**Membrane fractionation experiments**
Membrane fractionation was performed as previously described[71,72]. In brief, 200 ODs of cells were lysed in LBi (without detergent) as described above. After low-speed elution, crude membrane and soluble fractions were obtained by centrifugation for 1 h at 21'000 × g at 4 °C. The membrane pellets were resuspended in LBi + 1% LMNG + 0.1% CHS. Insoluble material was cleared by centrifugation at 13'000 × g for 20 min at 4 °C. The cleared extracts were snap frozen at −80 °C until being used for co-immunoprecipitation or denatured to be loaded into SDS gels.

**Co-immunoprecipitation**
For Fig. S3D, 100 ODs of cells were lysed in RIPA buffer as described above. Equalised protein amounts were added to prewashed magnetic beads and incubated for 3 h on a rotor at 4 °C. Beads were washed three times with RIPA buffer and eluted for 10 min at 95 °C in 2x NuPAGE LDS sample buffer containing 5% of β-mercaptoethanol. For all other experiments, 200 ODs of cells were lysed in LBi + 1% LMNG + 0.1% CHS as described above. Equalised protein amounts were added to prewashed magnetic beads and incubated for 3 h on a rotor at 4 °C. Beads were washed three times with LBi + 1% NP-40 and eluted for 15 min at 65 °C in 2x NuPAGE LDS sample buffer containing 5% of β-mercaptoethanol. The list of the magnetic beads used in this study is available in Table S3.

**Protein dephosphorylation**
200 ODs of cells were lysed in EDTA-free lysis buffer (50 mM Tris-HCl, pH 7.4, 200 mM NaCl) with protease inhibitors (EDTA-free protease inhibitor cocktail (Roche, 4693159001), 1.5 mM sodium orthovanadate, 10 mM sodium fluoride, 100 μM PMSF) and processed for membrane fractionation as described above. 80 units of alkaline phosphatase (Roche 11097075001) were added to 100 μg of total protein when indicated and all samples were incubated at 37 °C for 1 h, before being denatured and run in SDS-PAGE gels.

**Mass spectrometry**
**Sample preparation and protein digestion.** *miST (General affinity beads, on-beads digestion):* Samples were digested following a modified version of the iST method[125] (named miST method). 25 μl of miST lysis buffer (1% Sodium deoxycholate, 100 mM Tris pH 8.6, 10 mM DTT), were added to the beads. After mixing and dilution 1:1 (v:v) with H2O, samples were heated 5 min at 75 °C. Reduced disulfides were alkylated by adding 13 μL of 160 mM chloroacetamide (33 mM final) and incubating for 45 min at 25 °C in the dark. After digestion with 1.0 μg of Trypsin/LysC mix (Promega #V5073) for 2 h at 25 °C, sample supernatants were transferred to new tubes. To remove sodium deoxycholate, two sample volumes of isopropanol containing 1% TFA were added to the digests and the samples were desalted on a strong cation exchange (SCX) plate (Oasis MCX; Waters Corp., Milford, MA) by centrifugation. After washing with isopropanol/1%TFA, peptides were eluted in 200 μL of 60% MeCN, 39% water, 1% (v/v) ammonia and dried by centrifugal evaporation.

**Liquid chromatography-mass spectrometry analyses.** *TIMS-TOF DDA (Ultimate):* LC-MS/MS analyses were carried out on a TIMS-TOF Pro (Bruker, Bremen, Germany) mass spectrometer interfaced through a nanospray ion source ('captive spray') to an Ultimate 3000 RSLCnano HPLC system (Dionex). Peptides were separated on a reversed-phase custom packed 45 cm C18 column (75 µm ID, 100 Å, Reprosil Pur 1.9 µm particles, Dr. Maisch, Germany) at a flow rate of 250 nl/min with a 2–27% acetonitrile gradient in 93 min followed by a ramp to 45% in 15 min and to 90% in 5 min (total method time: 140 min, all solvents contained 0.1% formic acid). Data-dependent acquisition was carried out using a standard TIMS PASEF method[126] with ion accumulation for 100 ms for each of the survey MS1 scan and the TIMS-coupled MS2 scans. Duty cycle was kept at 100%. Up to 10 precursors were targeted per TIMS scan. Precursor isolation was done with a 2 or 3 m/z windows below or above m/z 800, respectively. The minimum threshold intensity for precursor selection was 2500. If the inclusion list allowed it, precursors were targeted more than once to reach a minimum target total intensity of 20'000. Collision energy was ramped linearly based uniquely on the 1/k0 values from 20 (at 1/k0 = 0.6) to 59 eV (at 1/k0 = 1.6). Total duration of a scan cycle, including one survey and 10 MS2 TIMS scans, was 1.16 s. Precursors could be targeted again in subsequent cycles if their signal increased by a factor of 4.0 or more. After selection in one cycle, precursors were excluded from further selection for 60 s. Mass resolution in all MS measurements was approximately 35'000.

**Data processing.** *MaxQuant (DDA):* Data files were analysed with MaxQuant 1.6.14.0[127] incorporating the Andromeda search engine[128]. Cysteine carbamidomethylation was selected as a fixed modification while methionine oxidation and protein N-terminal acetylation were specified as variable modifications. The sequence databases used for searching were the *S. pombe* reference proteome based on the UniProt database (www.uniprot.org, version of 1 February 2021, containing 5140 sequences) and a 'contaminant' database containing the most usual environmental contaminants and enzymes used for digestion (keratins, trypsin, etc). Mass tolerance was 4.5 ppm on precursors (after recalibration) and 20 ppm on MS/MS fragments. The 'match between runs' feature was not activated. Both peptide and protein identifications were filtered at 1% FDR relative to hits against a decoy database built by reversing protein sequences.

**Data analysis.** *Perseus (DDA, DIA, TMT):* All subsequent analyses were done with the Perseus software package (version 1.6.15.0)[129]. Contaminant proteins were removed and intensity iBAQ values[130] were log2-transformed and normalised based on the median. After assignment to groups, only proteins quantified in at least 3 samples of one group were kept. After missing values imputation (based on normal distribution using Perseus default parameters), t-tests were carried out among all conditions, with Benjamini-Hochberg FDR correction for multiple testing (adjusted p-value threshold <0.05). Imputed values were later removed. The differences in means obtained from the tests were used for 1D enrichment analysis on associated GO/KEGG annotations as described[131]. The enrichment analysis was also FDR-filtered (Benjamini-Hochberg, Q-val<0.02). The mass spectrometry proteomics data have been deposited to the ProteomeXchange Consortium via the PRIDE partner repository with the dataset identifier PXD068764

**Statistical analysis**
Two-tailed Student *t*-tests were performed to compare two groups in Figs. 2E, 6F and S2C, S2D; S5B. For comparisons of more than two groups, initial Brown-Forsythe tests were performed to assess for equality of group variances. One-way ANOVA with Tukey multiple comparison test was performed for Figs. 1D, F and 3G, 4D, 5G, 6G, H; S4D. Kruskal-Wallis with Dunn's multiple comparison test was

performed for Figs. 1H and 2C, 5F, 6B; S1C, S2A; S3A. A two-way ANOVA with Šídák multiple comparisons test was performed for Fig. 6D. For boxplots, the central line indicates the median, the bottom and top edges of the box indicate the 25th and 75th percentiles and the whiskers extend to the smallest and largest values. No statistical method was used to predetermine sample size. The experiments were not randomized and measurements were taken from distinct samples.

**Reporting summary**
Further information on research design is available in the Nature Portfolio Reporting Summary linked to this article.

## Data availability
Source data for all western blots and quantitative data are provided with this paper. Proteomic dataset is available in PRIDE (project PXD068764): https://www.ebi.ac.uk/pride/archive/projects/PXD068764. The imaging data that support the findings of this study are available from the corresponding author upon reasonable request. Source data are provided with this paper.

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

## Acknowledgements

We thank Prof Li-Lin Du (NIBS, Beijing, China) for the Pil1 plasmids, Dr Serge Pelet (UNIL, Lausanne, Switzerland) for dPSTR reagents, Dr Manfredo Quadroni (UNIL, Lausanne, Switzerland) and the UNIL Protein Analysis Facility for mass spectrometry, Dr Yoel Klug (Oxford University, UK) for the membrane extraction protocol, Prof Aleksandar Vještica (UNIL, Lausanne, Switzerland) for strains and comments on the manuscript and Prof Omaya Dudin (UNIGE, Geneva, Switzerland), Dr Fangfang Lu (Broad Institute, Cambridge, United States) and members of the Martin lab for comments on the manuscript. This work was supported by grants from the Swiss National Science Foundation (#176396 and # 191990) and the European Research Council (SexYeast) to S.G.M.

## Author contributions

Conceptualisation, B.S. and S.G.M.; methodology, B.S. and L.Me.(Laura Merlini); validation, B.S., L.Me., M.B., W. Li and M.B.; software: W.L.; formal analysis, B.S., L.Me. and W.L.; investigation, B.S., L.Me. and M.B.; resources, L.Mi.(Laetitia Michon), S.G-L.; writing—original draft, B.S.; writing—review and editing, B.S., L.Me. and S.G.M.; visualisation, B.S.; supervision, S.G.M.; funding acquisition, S.G.M.; project administration, B.S. and S.G.M.

## Competing interests

The authors declare no competing interests.
