## [Transparent Peer Review file · Nature Communications]

Phosphoregulation of the novel hemi-arrestin MAPK scaffold Sms1 prevents untimely mating

Corresponding Author: Professor Sophie Martin

Version 0:

Reviewer comments:

Reviewer #1

(Remarks to the Author)

This manuscript describes the characterization of the *S. pombe* protein Ste7 (renamed Sms1 in this study) as “the” scaffold protein for the *S. pombe* pheromone response MAP kinase cascade. In this context it is defined as formally equivalent to the Ste5 scaffold of the *S. cerevisiae* pathway, but although providing the same function, the *S. pombe* protein is structurally totally distinct.

The authors initially select the protein of interest, SPAC23E2.03c, from a pool of 235 candidates identified in a pull-down assay that identified interactors with the “specialized” adapter protein Ste4. It became the primary focus in part because previous studies had shown the protein, named Ste7, to be a critical component of the mating pathway. The current authors suggest the protein is poorly characterized, but the paper identifying the KO phenotype involves solid analysis. The current manuscript proposes renaming the gene Sms1 from Ste7 to prevent confusion with the MAPK of the *S. cerevisiae* pathway, but the field has already lost that fight (the SpSte4 adapter vs the ScSte4 G protein beta subunit is a clear example). Therefore, I think the name should remain Ste7 as it fully describes the phenotype (mutants are sterile) and has primacy in the literature, but in any event I hope the *S. pombe* nomenclature committee will be consulted prior to any publication defining this gene as Sms1.

Initial experiments confirm the mating failure of the mutant, the loss of MAPK phosphorylation, and loss of pheromone-induced gene activation. Subsequent analysis directed at the role of the Ste7/Sms1 element starts by confirming the interaction with Ste4 – the assay used to by these investigators to identify Ste7/Sms1 in the first place. All these experiments are solidly informative and competently done but are not really moving the story forward. If space is any issue, some (or all) of these results could be moved to supplementary figures.

The authors now investigate the characteristics of the association of the Ste7/Sms1 protein with the Ste4 adapter in more depth. Alpha-fold modeling suggests beta sheet augmentation could link the arrestin fold module of Ste7/Sms1 to the Ras domain of the Ste4 adaptor. Previous *S. cerevisiae* analysis of the equivalent part of the circuit has the Ras domain (of Ste50) linking to proteins like Opy2 and small GTPases while the SAM domain links to the SAM domain of the MAPKK Ste11. This puts the Ste7/Sms1 protein apparently in a similar position to membrane association elements like Cdc42 and Opy2. Direct test of the association between the arrestin fold region of Ste7/Sms1 and Ste4 involved an eisosome-defined association assay to investigate the requirement of the arrestin and RA domains in the formation of the protein complex. The wild-type domains bring the two fusion proteins together in the eisosomes; mutation of the RA domain or the arrestin fold region block the interaction. The data presentation of this experiment is incomplete – we are shown the GFP signal for the Ste7/Sms1 fusion protein, and the merge, but the mCherry signal for the Pil1-Ste4 fusion is not independently presented. This should be done – we need to see the structure of the signal for the eisosome-localized binding target. The functional analysis supports the picture provided by the structural study – constructs that are not associating do not allow normal signaling. The manuscript claims that these results support an association through a beta sheet augmentation, and that AlphaFold establishes that the interaction represents an example of protein association directing the folding of an unordered protein region. This is an over-interpretation of the data – the AlphaFold work is suggestive and not conclusive. It would be better to say the results are “consistent with” rather than “validate” the AlphaFold model.

At this point the Ste7/Sms1 protein has essentially been characterized as behaving like the Opy2 protein in *S. cerevisiae* – interacting with the RA domain of an adapter that also connects to an MPKK through its SAM domain. The next set of experiments investigates where the Ste7/Sms1 protein localizes in vegetative and signaling cells using GFP fusion constructs. The protein signal is increased under nitrogen starvation and further enhanced by pheromone treatment of the cells. Localization is in membrane patches, and ultimately sites of cell fusion in mating cells. The patches colocalize with Scd2, and are thus, along with Ste4, part of polarity patches in the membrane. Intriguingly, after cell fusion the fates of Ste4

and Ste7/Sms1 are different – with Ste4 remaining, and Ste7/Sms1 disappearing. When the structure of the protein was investigated in cell extracts the protein was found to be unstable and ubiquitinated – assessment of the influence of protein localization on stability found that the membrane associated version of the protein was both more stable and post-translationally modified. Nuclear localization of the GFP fusion protein was detected, but trapping the protein in the nucleus impacted protein stability and compromised mating, suggesting that nuclear export is essential for proper functioning, and supporting membrane localization as a key element in Ste7/Sms1 function.

To look directly at membrane association of the Ste7/Sms1 protein the authors mutate residues shown to be important for directing the structurally-similar mammalian beta arrestins to the plasma membrane. These modifications blocked cell surface localization of the *S. pombe* protein and caused sterility. Temperature sensitive inactivation of Its3, a PI4P convertase, also compromised Ste7/Sms1 cell surface localization. The structural similarities of the arrestin domains, and the involvement of phosphoinositides and lipid associating hydrophobic residues are taken to support a role for Ste7/Sms1 in interacting with the plasma membrane – this argument seems solid. Further assessment of the polarity-patch-linking element found that loss of the G protein alpha subunit blocked the polarization, while loss of Ras1 did not, leading the authors to propose the pheromone pathway G protein linkage was critical for the association Ste7/Sms1 into pheromone-induced-polarization and for mating. Additional testing suggests the linkage involves the activated form of the alpha subunit and requires the arrestin domain of the putative adapter protein.

While all these investigations provide convincing evidence of the Ste7/Sms1 protein playing a critical role in the pheromone pathway that involves directed membrane association and G protein binding, the authors needed to show critically-required-binding of the other pheromone pathway kinases to claim a general scaffold function for the protein. To investigate these bindings and their critical role in the pheromone response, they first looked into direct intracellular binding using immunoprecipitation in cells compromised for mating due to missing signaling components. CoIP shows interaction between the arrestin protein and both Byr1 and Spk1. This experiment is poorly described - as I understand it the Byr1 protein, which is mutated to be constitutively activated, is tagged with V5, but the legend says the protein was identified by the anti-FLAG antibody? The logic of the activation of Byr1 for the assay is not explained – why not use tagged wild type protein in the assay? These experiments are the most critical to establish the central point of the manuscript – the arrestin-like protein Ste7/Sms1 is a MAP kinase cascade scaffold. Therefore, the story here needs to be clear and unambiguous.

Further assessment of complex formation in mating competent cells involves FLAG-tagged Ste7/Sms1 co-immunoprecipitating phospho-Spk1 as assessed using a phospho-ERK antibody. MAP kinase coated beads could pull out full length Ste7/Sms1 from the membrane fraction of the cell. These assays don't ensure that all the proteins are complexed at the same time, but do provide solid evidence for interaction of the arrestin protein with all the kinase members of the mating kinase cascade.

Functional assays established that in the absence of the scaffold constitutively active MEK (activated Byr1) the MAPK Spk1 remained phosphorylated, but the constitutive shmooing phenotype characteristic of the activated MEK kinase is blocked, suggesting the role of the scaffold is not simply to facilitate the kinase phosphorylation cascade.

The final classes of experiments relate to negative feedback removing the arrestin protein from the membrane to keep mating off under non-mating-conducive conditions. This is shown to be phosphorylation controlled, and proteins mutated for “all” candidate MAP kinase Spk1 sites lack any phosphorylation-directed gel shifts. This non-phosphorylatable protein shows improper localization, and directs hyper response to pheromone, while generating pseudophosphorylated residues at the candidate phosphorylation sites abolished membrane localization. These experiments generally establish a role of phosphorylation, at least in part likely from Spk1, as playing an important role in Ste7/Sms1 function. More detailed analysis of site specificity and the role of other non-proline directed sites remain to be done.

At the end of the discussion the authors wonder if the scaffolds are evolving independently. That was the prediction for the Ste5/Far1 development in *C. albicans* - the Ste5 vWA domain was structurally distinct from the Far1 vWA domain, suggesting independent development, not simply duplication and divergence (ref 45).

Minor points

In the abstract and the start of the discussion the Sms1 protein is defined as a “structurally novel” scaffold. The arrestin fold is not “structurally novel”, and this protein, while representing a new scaffold, is not characterized in a way that highlights much about its structure being of dramatic interest. I think novel should not be used in this context.

The phrase “Because Sms1 binds the active G α -GTP at the plasma membrane and all MAPKs,” is awkward

Reference 45 is not discussing MAPK duplication, but rather the potential duplication of Far1 into the Far1/Ste5 pair.

“...spurious noise signal...” is awkward

Reviewer #2

(Remarks to the Author)

This very nice manuscript reports the identification and characterization of the previously unknown scaffold for the MAPK signaling cascade that controls mating in the fission yeast, *Schizosaccharomyces pombe*. The data convincingly demonstrate that the previously understudied protein Sms1 acts as the missing membrane-bound scaffold required for transducing the mating signal, and interacts with all components of the MAPK signaling pathway. The paper further describes how this new scaffold is regulated – by upregulation during mating, nuclear export, lipid binding, partner binding, ubiquitination, and negative feedback via phosphorylation. In my opinion, it is a very comprehensive work with no significant weaknesses. The data are strong and support the conclusions. The paper is also well-written and will be of wide interest to the protein kinase signaling field.

I do have some suggestions for changes to the text/figures. These are all relatively minor and listed below.

Abstract:

a) Add an "a" to "scaffolds that recruit kinases to subcellular locations and promote signal transduction have only been described in few species" between in and few.

Intro and discussion:

a) Please add a "the" between in and presence "even in presence of hyperactive MAP2K"

b) Rather short shrift is given to the fact that Sms1 was previously known as Ste7 and some information was known about Ste7. I think this information should be included in the final paragraph of the introduction in addition to mentioning it in the results section.

c) The discussion would also benefit from describing how the new information regarding Sms1/Ste7 function can explain/expand on or contrasts with what was previously known about Ste7.

Results:

1) "Amongst Ste4 interactors, we focused on the poorly characterised protein SPAC23E2.03c (UniprotKB Q10136), previously known as Ste7

and shown to promote mating and meiosis." I recommend replacing "and" in this sentence with "because it was previously".

2) A more detailed explanation for why the observed results in Fig S3 are due to Ste11-dependent transcription would be helpful.

3) "Whole-cell extraction of mating cells expressing Sms1-sfGFP only showed low molecular weight fragments". This sentence confused me for a couple of reasons. First, low molecular weight fragments were observed by immunoblotting of whole cell lysates, not by extraction. Second, the sentence/paragraph would benefit from further explanation and contrasting the conditions with previous co-IP experiments/blots shown earlier and blots shown later (e.g. Fig. 3E) in the paper. These other blots showed more than low molecular weight fragments. Expanding the explanation of exactly what the differences in conditions are would avoid confusion and promote a better understanding of these variable blotting results.

4) "differentially regulated at the membrane (polarity patches and fusion site) and the cytosol." The meaning would be clearer if the second "and" in this sentence were replaced with "versus".

5) In "its restriction at the polarity patches and fusion site", please replace "at" with "to".

6) The sentence "By contrast, Sms1 polarisation was abolished in absence of the GαGpa1 (Fig. 4E)" seems to be in the wrong place in the text. For a more logical flow and better understanding, I suggest re-working this sentence and those in the following paragraph.

7) The preferential binding of Sms1 to the GTP-locked version of Gpa1 is difficult to substantiate without quantification of the blots.

8) "loss of Sms1 only moderately decreased pSpk1 level (Fig. 5E)". This statement also requires quantification of the blots.

9) "the phospho-binding proteins 14-3-3, Rad24"; please delete "s" from proteins

Figure/figure legends:

1) In figure 1, inconsistency with Sexual capitalized and cell not.

2) Throughout the figures, tubulin is used as a control for the amount of soluble protein but included elsewhere (although not always) as if it is a loading control for the membrane fractions (example Fig. 1C) and of course, it is not a loading control for the membrane fractions. Therefore, it seems that the blue banner of "soluble" should be added above all the tubulin blots and/or an explanation of this control and its appearance under membrane fraction blots should be provided.

3) In the legend for Fig. 1E, it says the volcano plot represents the results of immunoprecipitations. This description is incomplete because it is a representation of the MS results and this could be expanded.

4) Legend to figures 1H and S1C should include the number of cells analyzed – looks like many cells were analyzed.

5) Legend to Figure 4, last sentence should end in controls rather than control.

6) The input and IP panels of Fig. 5C. are not well-aligned in that pSpk1 bands in the two panels are at different distances from the markers.

7) Please clarify whether or not the Sms1 blots are from lysates or IPs in Fig 6A.

Methods:

1. Yeast media composition and standard methods should be referenced.

2. Please describe constructs for the eisosome experiments.

Reviewer #3

(Remarks to the Author)

Review of Negative feedback regulation of the hemi-arrestin MAPK scaffold Sms1 prevents untimely mating, by Boris Sieber et al.

In this manuscript the authors make the surprising discovery of a scaffold protein of the Sch. pombe pheromone response pathway. After 30 years of investigation in which the community thought there was no scaffold, the group of Dr. Sophie Martin present clear evidence that the protein they rebaptized Sms1 (formerly known as SPAC23E2.03c or Ste7) is actually an essential component working as a scaffold.

They show convincing evidence all along the manuscript.

Sms1 binds the Galpha subunit Gpa1, the cofactor Ste4 (which brings along the Map3K Byr2), the MapK2 Byr1 and the

MapK Spk1.

Sms1 associates with the lipids of the plasma membrane, and both interactions (Gpa1 and lipids) are necessary for Sms1 to be recruited to the PM. Its association to the PM stabilizes it and is essential for shmoo formation.

Sms1 seems to be negatively regulated by multiple S/T-P phosphorylations, since the 22A mutant is hyperactive and the 22E is not active and cannot associate with the PM and they suggest it is negative feedback from the MAPK.

Although Spk1 needs Sms1 localization to be activated by Byr1, Sms1 is not necessary if Byr1 is constitutively active; that is, Sms1 is not an allosteric modulator of Spk1.

This is a remarkable paper, with data well supported by multiple experiments. It is quite amazing that Sms1 performs almost all the functions and operates quite similarly to the budding yeast Ste5 scaffold, but it shares no common evolutionary origin with it.

Main comments:

Comment 1- There is no direct evidence that Sms1 is regulated by Spk1-dependent negative feedback. The strong evidence is that it is regulated by phosphorylation in S/T-P sites, but experiments are needed to establish that Spk1 is the kinase.

Given the sheer number of new results, I'm somewhat puzzled that this particular aspect was chosen as the main result of the paper, appearing in the title itself.

The current evidence is this:

1- The 22A Sms1 mutant yeast show a hyperactive pathway and Sms1 highly localized at the membrane. The exact opposite happens to the 22E Sms1 yeast. This means that phosphorylation of Sms1 negatively regulates it.

2- Deletion of Spk1 causes a similar but milder phenotype as the 22A Sms1 mutant.

That is all. To establish that it is a negative feedback one would need results of this sort (these are only examples):

a) Deletion of Spk1 results in complete loss (or extensive reduction) of the high MW bands that are shown in Fig6A

b) Inhibition of Spk1-as mutants with PP1 analogs causes spontaneous recruitment of Sms1 to the membrane

c) Byr1-dd suppresses Sms1 PM localization in a Spk1 dependent fashion

d) In vitro phosphorylation assays that show that Spk1 can phosphorylate Sms1 (or fragments of it).

I don't think this sort of experiments need to be done for the paper to be accepted, but to claim that Spk1 acts on Sms1 in a negative feedback loop, yes. So, I strongly encourage the authors to change the title accordingly.

Comment 2. It is not clear if they tested the direct interaction between Sms1 and Byr2. Please add that information.

Version 1:

Reviewer comments:

Reviewer #1

(Remarks to the Author)

The corrections made in the manuscript address my concerns adequately.

Reviewer #2

(Remarks to the Author)

The majority of my comments have been addressed satisfactorily. However, I found the level of response to three points a bit perfunctory and therefore insufficient. Specifically, I am not yet satisfied with how previous knowledge regarding Ste7 is introduced in the introduction. William Wells, a JCB editor, wrote a piece regarding a rising tendency (way back in 2006) to put what should be in an introduction into the discussion, sometimes in an effort to make work seem more novel than it is. To avoid this interpretation of motive by others, I refer the authors to PMID: 16954351 and ask that they take the recommendations in this perspective more seriously. What was known previously about Ste7 should be in the introduction to help the readers put the new work in context. This would have helped me better appreciate the conceptual advance of the present work. I also still think that the reason effort was put into Sms1/Ste7 from the list of proteomics hits was BECAUSE of what was known previously about this gene product. Finally, I didn't really understand still from a couple of comments in the discussion what was known previously about Ste7.

Response to reviewers

Reviewer 1

This manuscript describes the characterization of the *S. pombe* protein Ste7 (renamed Sms1 in this study) as “the” scaffold protein for the *S. pombe* pheromone response MAP kinase cascade. In this context it is defined as formally equivalent to the Ste5 scaffold of the *S. cerevisiae* pathway, but although providing the same function, the *S. pombe* protein is structurally totally distinct.

The authors initially select the protein of interest, SPAC23E2.03c, from a pool of 235 candidates identified in a pull-down assay that identified interactors with the “specialized” adapter protein Ste4. It became the primary focus in part because previous studies had shown the protein, named Ste7, to be a critical component of the mating pathway. The current authors suggest the protein is poorly characterized, but the paper identifying the KO phenotype involves solid analysis. The current manuscript proposes renaming the gene Sms1 from Ste7 to prevent confusion with the MAPK of the *S. cerevisiae* pathway, but the field has already lost that fight (the SpSte4 adapter vs the ScSte4 G protein beta subunit is a clear example). Therefore, I think the name should remain Ste7 as it fully describes the phenotype (mutants are sterile) and has primacy in the literature, but in any event I hope the *S. pombe* nomenclature committee will be consulted prior to any publication defining this gene as Sms1.

The reviewer can be reassured. The name change was performed with agreement from the first author of the original Ste7 paper and received the approval of the PomBase Gene naming committee (<https://www.pombase.org/submit-data/gene-naming-committee-members>). Upon their validation, the gene name has already been changed on PomBase (<https://www.pombase.org/gene/SPAC23E2.03c>) with Ste7 appearing as a synonym. We have now specified in the manuscript that the name change was approved by PomBase.

Initial experiments confirm the mating failure of the mutant, the loss of MAPK phosphorylation, and loss of pheromone-induced gene activation. Subsequent analysis directed at the role of the Ste7/Sms1 element starts by confirming the interaction with Ste4 – the assay used to by these investigators to identify Ste7/Sms1 in the first place. All these experiments are solidly informative and competently done but are not really moving the story forward. If space is any issue, some (or all) of these results could be moved to supplementary figures. The authors now investigate the characteristics of the association of the Ste7/Sms1 protein with the Ste4 adapter in more depth. Alpha-fold modeling suggests beta sheet augmentation could link the arrestin fold module of Ste7/Sms1 to the Ras domain of the Ste4 adaptor. Previous *S. cerevisiae* analysis of the equivalent part of the circuit has the Ras domain (of Ste50) linking to proteins like Opy2 and small GTPases while the SAM domain links to the SAM domain of the MAPKK Ste11. This puts the Ste7/Sms1 protein apparently in a similar position to membrane association elements like Cdc42 and Opy2. Direct test of the association between the arrestin fold region of Ste7/Sms1 and Ste4 involved an eisosome-defined association assay to investigate the requirement of the arrestin and RA domains in the formation of the protein complex. The wild-type domains bring the two fusion proteins together in the eisosomes; mutation of the RA domain

or the arrestin fold region block the interaction. The data presentation of this experiment is incomplete – we are shown the GFP signal for the Ste7/Sms1 fusion protein, and the merge, but the mCherry signal for the Pil1-Ste4 fusion is not independently presented. This should be done – we need to see the structure of the signal for the eisosome-localized binding target.

We have added the pictures of the mCherry channel in Figure 2C and S2B as suggested.

The functional analysis supports the picture provided by the structural study – constructs that are not associating do not allow normal signaling. The manuscript claims that these results support an association through a beta sheet augmentation, and that AlphaFold establishes that the interaction represents an example of protein association directing the folding of an unordered protein region. This is an over-interpretation of the data – the AlphaFold work is suggestive and not conclusive. It would be better to say the results are “consistent with” rather than “validate” the AlphaFold model.

We have changed the wording as suggested.

At this point the Ste7/Sms1 protein has essentially been characterized as behaving like the Opy2 protein in *S. cerevisiae* – interacting with the RA domain of an adapter that also connects to an MPKK through its SAM domain. The next set of experiments investigates where the Ste7/Sms1 protein localizes in vegetative and signaling cells using GFP fusion constructs. The protein signal is increased under nitrogen starvation and further enhanced by pheromone treatment of the cells. Localization is in membrane patches, and ultimately sites of cell fusion in mating cells. The patches colocalize with Scd2, and are thus, along with Ste4, part of polarity patches in the membrane. Intriguingly, after cell fusion the fates of Ste4 and Ste7/Sms1 are different – with Ste4 remaining, and Ste7/Sms1 disappearing. When the structure of the protein was investigated in cell extracts the protein was found to be unstable and ubiquitinated – assessment of the influence of protein localization on stability found that the membrane associated version of the protein was both more stable and post-translationally modified. Nuclear localization of the GFP fusion protein was detected, but trapping the protein in the nucleus impacted protein stability and compromised mating, suggesting that nuclear export is essential for proper functioning, and supporting membrane localization as a key element in Ste7/Sms1 function. To look directly at membrane association of the Ste7/Sms1 protein the authors mutate residues shown to be important for directing the structurally-similar mammalian beta arrestins to the plasma membrane. These modifications blocked cell surface localization of the *S. pombe* protein and caused sterility. Temperature sensitive inactivation of Its3, a PI4P convertase, also compromised Ste7/Sms1 cell surface localization. The structural similarities of the arrestin domains, and the involvement of phosphoinositides and lipid associating hydrophobic residues are taken to support a role for Ste7/Sms1 in interacting with the plasma membrane – this argument seems solid. Further assessment of the polarity-patch-linking element found that loss of the G protein alpha subunit blocked the polarization, while loss of Ras1 did not, leading the authors to propose the pheromone pathway G protein linkage was critical for the association Ste7/Sms1 into pheromone-induced-polarization and for mating. Additional testing suggests the linkage involves the activated form of the alpha subunit and requires the arrestin domain of the putative adapter protein.

While all these investigations provide convincing evidence of the Ste7/Sms1 protein playing a critical role in the pheromone pathway that involves directed membrane association and G protein

binding, the authors needed to show critically-required-binding of the other pheromone pathway kinases to claim a general scaffold function for the protein. To investigate these bindings and their critical role in the pheromone response, they first looked into direct intracellular binding using immunoprecipitation in cells compromised for mating due to missing signaling components. CoIP shows interaction between the arrestin protein and both Byr1 and Spk1. This experiment is poorly described - as I understand it the Byr1 protein, which is mutated to be constitutively activated, is tagged with V5, but the legend says the protein was identified by the anti-FLAG antibody?

We apologize for the confusion. The coIP was indeed done with V5-tagged constitutively active Byr1DD and immunoblotted with anti-V5 antibodies. The FLAG was a typo and has now been corrected with V5 in the figure legend.

The logic of the activation of Byr1 for the assay is not explained – why not use tagged wild type protein in the assay?

We performed the experiment with both wildtype Byr1 and Byr1DD. The interaction is clear, though seems a bit weaker with wildtype Byr1. We have now added the coIP of Sms1 with wildtype Byr1 as Figure S5A.

These experiments are the most critical to establish the central point of the manuscript – the arrestin-like protein Ste7/Sms1 is a MAP kinase cascade scaffold. Therefore, the story here needs to be clear and unambiguous. Further assessment of complex formation in mating competent cells involves FLAG-tagged Ste7/Sms1 co-immunoprecipitating phospho-Spk1 as assessed using a phospho-ERK antibody. MAP kinase coated beads could pull out full length Ste7/Sms1 from the membrane fraction of the cell. These assays don't ensure that all the proteins are complexed at the same time, but do provide solid evidence for interaction of the arrestin protein with all the kinase members of the mating kinase cascade. Functional assays established that in the absence of the scaffold constitutively active MEK (activated Byr1) the MAPK Spk1 remained phosphorylated, but the constitutive shmooing phenotype characteristic of the activated MEK kinase is blocked, suggesting the role of the scaffold is not simply to facilitate the kinase phosphorylation cascade. The final classes of experiments relate to negative feedback removing the arrestin protein from the membrane to keep mating off under non-mating-conducive conditions. This is shown to be phosphorylation controlled, and proteins mutated for "all" candidate MAP kinase Spk1 sites lack any phosphorylation-directed gel shifts. This non-phosphorylatable protein shows improper localization, and directs hyper response to pheromone, while generating pseudophosphorylated residues at the candidate phosphorylation sites abolished membrane localization. These experiments generally establish a role of phosphorylation, at least in part likely from Spk1, as playing an important role in Ste7/Sms1 function. More detailed analysis of site specificity and the role of other non-proline directed sites remain to be done.

Indeed, site specificity, and the possible role of other proline-directed kinases in regulating Sms1 will be probed in future studies.

At the end of the discussion the authors wonder if the scaffolds are evolving independently. That

was the prediction for the Ste5/Far1 development in *C. albicans* - the Ste5 vWA domain was structurally distinct from the Far1 vWA domain, suggesting independent development, not simply duplication and divergence (ref 45).

The case here is a different from the Ste5/Far1 example. The domain organisation of Ste5 and Far1 is very similar, though their vWA are structurally distinct. The cited study examined the evolution of these proteins and showed that emergence of Ste5 predates the whole-genome duplication event in *Saccharomycotina*, but still hypothesized it arose by duplication of part of the Far1 gene. By contrast, Sms1 is structurally completely distinct from Ste5, and we discuss how evolutionary forces may have led to co-option of distinct proteins that converge to similar scaffolding functions.

Minor points

In the abstract and the start of the discussion the Sms1 protein is defined as a “structurally novel” scaffold. The arrestin fold is not “structurally novel”, and this protein, while representing a new scaffold, is not characterized in a way that highlights much about its structure being of dramatic interest. I think novel should not be used in this context.

We agree that the arrestin fold is not per se novel. The function of the protein as a scaffold is. We changed the wording from “structurally novel scaffold” to “novel scaffold”. We point out the distinct structural fold at the end of the abstract.

The phrase “Because Sms1 binds the active G α -GTP at the plasma membrane and all MAPKs,” is awkward

Agreed. We changed the sentence to “Because Sms1 binds G α and all MAPKs at the plasma membrane,…”

Reference 45 is not discussing MAPK duplication, but rather the potential duplication of Far1 into the Far1/Ste5 pair.

While the main message of that paper is indeed on the Far1/Ste5 function, the phylogenetic analysis shows that Ste5 arose simultaneously with the MAPK duplication observed in *Saccharomycotina*, so we feel this is an appropriate citation.

“...spurious noise signal...” is awkward

We have changed this wording to “...spurious environmental noise...”

Reviewer 2

This very nice manuscript reports the identification and characterization of the previously unknown scaffold for the MAPK signaling cascade that controls mating in the fission yeast, *Schizosaccharomyces pombe*. The data convincingly demonstrate that the previously understudied protein Sms1 acts as the missing membrane-bound scaffold required for transducing the mating signal, and interacts with all components of the MAPK signaling pathway. The paper further describes how this new scaffold is regulated – by upregulation during mating, nuclear export, lipid binding, partner binding, ubiquitination, and negative feedback via phosphorylation. In my opinion, it is a very comprehensive work with no significant weaknesses. The data are strong and support the conclusions. The paper is also well-written and will be of wide interest to the protein kinase signaling field.

I do have some suggestions for changes to the text/figures. These are all relatively minor and listed below.

Abstract:

a) Add an “a” to “scaffolds that recruit kinases to subcellular locations and promote signal transduction have only been described in few species” between in and few.

Done

Intro and discussion:

a) Please add a "the" between in and presence "even in presence of hyperactive MAP2K"

Done

b) Rather short shrift is given to the fact that Sms1 was previously known as Ste7 and some information was known about Ste7. I think this information should be included in the final paragraph of the introduction in addition to mentioning it in the results section.

c) The discussion would also benefit from describing how the new information regarding Sms1/Ste7 function can explain/expand on or contrasts with what was previously known about Ste7.

We agree. We have added mention that Sms1 was previously known as Ste7 at the end of the introduction and highlighted that Sms1 was known both for its role in cell conjugation and in meiosis repression. We have also added in the discussion that Pat1, found as an Sms1 interactor by (Matsuyama et al., 2000), represents an interesting candidate for repression of Sms1 in nitrogen-rich conditions.

Results:

1) “Amongst Ste4 interactors, we focused on the poorly characterised protein SPAC23E2.03c (UniprotKB Q10136), previously known as Ste7 and shown to promote mating and meiosis.” I recommend replacing “and” in this sentence with “because it was previously”.

We have changed that sentence.

2) A more detailed explanation for why the observed results in Fig S3 are due to Ste11-dependent transcription would be helpful.

Ste11 is de-repressed upon nitrogen starvation and further enhanced upon pheromone stimulation. Previous -omics studies had also shown that sms1 transcription is Ste11-dependent. We have rephrased the text to make it clearer that it is an inference from our data, consistent with previously published literature.

3) “Whole-cell extraction of mating cells expressing Sms1-sfGFP only showed low molecular weight fragments”. This sentence confused me for a couple of reasons. First, low molecular weight fragments were observed by immunoblotting of whole cell lysates, not by extraction. Second, the sentence/paragraph would benefit from further explanation and contrasting the conditions with previous co-IP experiments/blots shown earlier and blots shown later (e.g. Fig. 3E) in the paper. These other blots showed more than low molecular weight fragments. Expanding the explanation of exactly what the differences in conditions are would avoid confusion and promote a better understanding of these variable blotting results.

The blots shown in Fig S3C and S3D were obtained before optimisation of the protein extraction protocol. In these early experiments, we could not recover any full-length Sms1 protein. We then tested multiple parameters in the extraction buffer and protocol to attempt to improve recovery of full-length protein. The main important change was to perform membrane extraction but change of the detergent also helped. In the early experiments, we used RIPA buffer, which contains SDS and NP-40. We then substituted these detergents for LMNG. All other experiments were performed in these improved solubilization conditions.

We have added a sentence in the main text to explain this optimization, which refers to the methods, where this is described in detail: “Optimization of the extraction buffer and protocol (see methods) allowed to recover some full-length Sms1, though a large fraction was still degraded (Fig. 3E, soluble fraction).”

4) “differentially regulated at the membrane (polarity patches and fusion site) and the cytosol.” The meaning would be clearer if the second “and” in this sentence were replaced with “versus”.

Done

5) In “its restriction at the polarity patches and fusion site”, please replace “at” with “to”.

Done

6) The sentence “ By contrast, Sms1 polarisation was abolished in absence of the GαGpa1 (Fig. 4E)” seems to be in the wrong place in the text. For a more logical flow and better understanding, I suggest re-working this sentence and those in the following paragraph.

We have moved this information to the next paragraph as suggested.

7) The preferential binding of Sms1 to the GTP-locked version of Gpa1 is difficult to substantiate without quantification of the blots.

Thank you for noticing this omission. Upon quantification of the experimental replicates, it became apparent that there is too much variability to conclude on a preferential binding of Sms1 for GTP-locked Gpa1. The quantification of the three independent experiments is now added as Fig. S4D. We also modified the manuscript accordingly by only mentioning the interaction of Sms1 with Gpa1, irrespective of its activation status.

8) “loss of Sms1 only moderately decreased pSpk1 level (Fig. 5E)”. This statement also requires quantification of the blots.

We have modified this statement as the loss of Sms1 does not significantly affect pSpk1 level in the quantification of the three biological replicates (added as Fig. S5B).

9) “the phospho-binding proteins 14-3-3, Rad24”; please delete “s” from proteins

Done

Figure/figure legends:

1) In figure 1, inconsistency with Sexual capitalized and cell not.

Corrected

2) Throughout the figures, tubulin is used as a control for the amount of soluble protein but included elsewhere (although not always) as if it is a loading control for the membrane fractions (example Fig. 1C) and of course, it is not a loading control for the membrane fractions. Therefore, it seems that the blue banner of “soluble” should be added above all the tubulin blots and/or an explanation of this control and its appearance under membrane fraction blots should be provided.

Indeed, the tubulin was detected in the soluble fraction. We have added to the figure: tubulin (soluble).

3) In the legend for Fig. 1E, it says the volcano plot represents the results of immunoprecipitations. This description is incomplete because it is a representation of the MS results and this could be expanded.

The legend has been expanded to be more precise.

4) Legend to figures 1H and S1C should include the number of cells analyzed – looks like many cells were analyzed.

N values are now provided in the legend

5) Legend to Figure 4, last sentence should end in controls rather than control.

Corrected

6) The input and IP panels of Fig. 5C. are not well-aligned in that pSpk1 bands in the two panels are at different distances from the markers.

Thank for noticing this alignment issue, we have now adjusted the orientation of the phospho-ERK1/2 IP panel to ensure alignment of the pSpk1 band.

7) Please clarify whether or not the Sms1 blots are from lysates or IPs in Fig 6A.

These are from lysates. The information is now in the legend.

Methods:

1. Yeast media composition and standard methods should be referenced.

We have added reference for the composition of the MSL medium, which is the critical one for this study.

2. Please describe constructs for the eisosome experiments.

Thanks for spotting this omission. The constructs and experiments are now described in the methods.

Reviewer #3:

In this manuscript the authors make the surprising discovery of a scaffold protein of the *Sch. pombe* pheromone response pathway. After 30 years of investigation in which the community thought there was no scaffold, the group of Dr. Sophie Martin present clear evidence that the protein they rebaptized Sms1 (formerly known as SPAC23E2.03c or Ste7) is actually an essential component working as a scaffold. They show convincing evidence all along the manuscript. Sms1 binds the Galpha subunit Gpa1, the cofactor Ste4 (which brings along the Map3K Byr2), the MapK2 Byr1 and the MapK Spk1. Sms1 associates with the lipids of the plasma membrane, and both interactions (Gpa1 and lipids) are necessary for Sms1 to be recruited to the PM. It's association to the PM stabilizes it and is essential for shmoo formation. Sms1 seems to be negatively regulated by multiple S/T-P phosphorylations, since the 22A mutant is hyperactive and the 22E is not active and cannot associate with the PM and they suggest it is negative feedback from the MAPK. Although Spk1 needs Sms1 localization to be activated by Byr1, Sms1 is not necessary if Byr1 is constitutively active; that is, Sms1 is not an allosteric modulator of Spk1. This is a remarkable paper, with data well supported by multiple experiments. It is quite amazing that Sms1 performs almost all the functions and operates quite similarly to the budding yeast Ste5 scaffold, but it shares no common evolutionary origin with it.

Main comments:

Comment 1- There is no direct evidence that Sms1 is regulated by Spk1-dependent negative feedback. The strong evidence is that it is regulated by phosphorylation in S/T-P sites, but experiments are needed to establish that Spk1 is the kinase. Given the sheer number of new results, I'm somewhat puzzled that this particular aspect was chosen as the main result of the paper, appearing in the title itself. The current evidence is this: 1- The 22A Sms1 mutant yeast show a hyperactive pathway and Sms1 highly localized at the membrane. The exact opposite happens to the 22E Sms1 yeast. This means that phosphorylation of Sms1 negatively regulates it. 2- Deletion of Spk1 causes a similar but milder phenotype as the 22A Sms1 mutant. That is all. To establish that it is a negative feedback one would need results of this sort (these are only examples): a) Deletion of Spk1 results in complete loss (or extensive reduction) of the high MW bands that are shown in Fig6A b) Inhibition of Spk1-as mutants with PP1 analogs causes spontaneous recruitment of Sms1 to the membrane c) Byr1-dd suppresses Sms1 PM localization in a Spk1 dependent fashion d) In vitro phosphorylation assays that show that Spk1 can phosphorylate Sms1 (or fragments of it).

I don't think this sort of experiments need to be done for the paper to be accepted, but to claim that Spk1 acts on Sms1 in a negative feedback loop, yes. So, I strongly encourage the authors to change the title accordingly.

We agree with this comment. Although it is very likely that Spk1 functions in negative feedback regulation of Sms1, the experiments we have done so far do not directly demonstrate this point. We have changed the title accordingly. Future work will address whether Spk1 indeed directly phosphorylates Sms1.

Comment 2. It is not clear if they tested the direct interaction between Sms1 and Byr2. Please add that information.

We have not tested whether Sms1 directly binds Byr2. This is now mentioned in the discussion.

Response to reviewers

Reviewer #2:

The majority of my comments have been addressed satisfactorily. However, I found the level of response to three points a bit perfunctory and therefore insufficient. Specifically, I am not yet satisfied with how previous knowledge regarding Ste7 is introduced in the introduction. William Wells, a JCB editor, wrote a piece regarding a rising tendency (way back in 2006) to put what should be in an introduction into the discussion, sometimes in an effort to make work seem more novel than it is. To avoid this interpretation of motive by others, I refer the authors to PMID: 16954351 and ask that they take the recommendations in this perspective more seriously. What was known previously about Ste7 should be in the introduction to help the readers put the new work in context. This would have helped me better appreciate the conceptual advance of the present work. I also still think that the reason effort was put into Sms1/Ste7 from the list of proteomics hits was BECAUSE of what was known previously about this gene product. Finally, I didn't really understand still from a couple of comments in the discussion what was known previously about Ste7.

The reviewer is right that the known sterility of *sms1/ste7Δ* oriented our decision to investigate this protein in more depth, and we have never hidden this information. The only previous work on Sms1/Ste7 cloned the gene, thus attributing molecular information to the sterility phenotype, and showed that Sms1/Ste7 inhibits meiosis upon overexpression. It further showed that the protein is unstable in zygotes and that it interacts (in two-hybrid assays) with the RNA-binding protein Mei2, the master regulator of meiosis, and with its Pat1 kinase inhibitor.

We have introduced the previous knowledge on the sterility of *sms1/ste7Δ* in the introduction and have now slightly expanded on it. We then bring the further information on Sms1/Ste7 at the time we first describe it in the results, rather than in the introduction. This is not to hide information, but to make the flow clearer. Sms1 was discovered in our biochemical screen, the setup of which was not based on any knowledge about Sms1/Ste7. Indeed, the importance of the previous work on Sms1/Ste7 was clearly not appreciated by the field (and we have to admit, we were not aware of it before encountering the protein). The Matsuyama et al paper has been cited 11 times since 2000, 6 times for technical points (not even mentioning Sms1/Ste7), 3 times by listing Sms1/Ste7 as an example of an arrestin domain-containing protein, once as mis-citation of Ste7 MAP2K and only once for the biology it reported. In no way does this reduce its importance, but clearly the work was not a centre piece of our understanding of the yeast mating process.

We have also added a sentence in the discussion on the role of its degradation in zygotes, as additional regulation to the phosphorylation we propose, to promote zygotic events. It is difficult to write more, as there is currently no understanding on how the reported interactions with Mei2 and Pat1 may link or not with its role as pheromone-MAPK scaffold.